
# Satellite observations of middle atmosphere gravity wave activity and dissipation during recent stratospheric warmings

Manfred Ern[1], Quang Thai Trinh[1], Martin Kaufmann[1], Isabell Krisch[1], Peter Preusse[1], Jörn Ungermann[1], Yajun Zhu[1], John C. Gille[2,3], Martin G. Mlynczak[4], James M. Russell III[5], Michael J. Schwartz[6], and Martin Riese[1]

[1]Institut für Energie- und Klimaforschung – Stratosphäre (IEK–7), Forschungszentrum Jülich GmbH, 52425 Jülich, Germany
[2]Center for Limb Atmospheric Sounding, University of Colorado at Boulder, Boulder, Colorado, USA.
[3]National Center for Atmospheric Research, Boulder, Colorado, USA.
[4]NASA Langley Research Center, Hampton, Virginia, USA.
[5]Center for Atmospheric Sciences, Hampton University, Hampton, Virginia, USA.
[6]Jet Propulsion Laboratory, California Institute of Technology, Pasadena, California, USA.

*Correspondence to:* M. Ern (m.ern@fz-juelich.de)

**Abstract.** Sudden stratospheric warmings (SSWs) are circulation anomalies in the polar region during winter. They mostly occur in the Northern Hemisphere and affect also surface weather and climate. Both planetary waves and gravity waves contribute to the onset and evolution of SSWs. While the role of planetary waves for SSW evolution has been recognized, the effect of gravity waves is still not fully understood, and has not been comprehensively analyzed based on global observations.

5   In particular, information on the gravity wave driving of the background winds during SSWs is still missing.

We investigate the boreal winters 2001/2002 until 2013/2014. Absolute gravity wave momentum fluxes and gravity wave dissipation (potential drag) are estimated from temperature observations of the satellite instruments HIRDLS and SABER. In agreement with previous work, we find that sometimes gravity wave activity is enhanced before the central date of major SSWs, particularly during vortex-split events. Often, SSWs are associated with polar-night jet oscillation (PJO) events. For

10   these events, we find that gravity wave activity is strongly suppressed when winds reverse from eastward to westward (usually after the central date of a major SSW). In addition, gravity wave potential drag at the bottom of the newly forming eastward directed jet is remarkably weak, while considerable potential drag at the top of the jet likely contributes to the downward propagation of both the jet and the new elevated stratopause. During PJO events, we also find some indication for poleward propagation of gravity waves. Another striking finding is that obviously localized gravity wave sources, likely mountain waves

15   and jet-generated gravity waves, play an important role during the evolution of SSWs and potentially contribute to the triggering of SSWs by preconditioning the shape of the polar vortex. The distribution of these hot spots is highly variable and strongly depends on the zonal and meridional shape of the background wind field, indicating that a pure zonal average view sometimes is a too strong simplification for the strongly perturbed conditions during the evolution of SSWs.





## 1 Introduction

The unperturbed arctic winter stratosphere is characterized by a strong eastward directed zonal wind jet (polar vortex). Occasionally, however, forcing by upward propagating planetary Rossby waves can lead to strong deceleration and even reversals of this polar jet. These events are associated with strong warming of the polar stratosphere, and they are therefore called sudden
stratospheric warmings (SSWs). Such events were first reported by Scherhag (1952), and the importance of upward propagating planetary waves for the driving of SSWs was first pointed out by Matsuno (1971). A climatology and characterization of SSW events into different categories can be found, for example, in Charlton and Polvani (2007). Often, the following classification is used: During a "minor warming", the temperature gradient in the stratosphere between 60° latitude and pole, on zonal average, becomes positive over a certain altitude range at or below the 10 hPa level (about 32 km altitude). During a "major warming",
additionally, the stratospheric zonal wind at 60° latitude reverses from eastward to westward at or below the 10 hPa level (e.g., Chandran et al., 2014).

SSWs are dynamical processes that involve strong vertical coupling through atmospheric waves. Much of the dynamics during SSWs can be understood by upward propagation of planetary waves from the troposphere, amplification of their amplitudes, followed by wave dissipation and heat flux convergence. During this process considerable wave drag is exerted on
the zonal mean background flow. Particularly during major warmings, stationary planetary waves will encounter critical levels when the zonal jet reverses from eastward to westward. The waves dissipate and can no longer propagate to higher altitudes. However, as was shown by Holton (1983), accurate representation of major SSWs also requires the inclusion of gravity wave drag in models.

### 1.1 Relevance of SSWs for global modeling

Simulating realistic SSWs in general circulation models and chemistry climate models (GCMs/CCMs) and including the physical processes that are involved in the onset and evolution of SSWs is important for several reasons:

(1) Effects of SSWs are not limited to the polar region. SSWs influence the global meridional residual circulation, and meridional coupling between different latitudes is observed. For example, SSWs have influence on mesospheric temperatures in the tropics (e.g., Shepherd et al., 2007), and they likely also have an effect on the opposite hemisphere (e.g., Becker and
Fritts, 2006; de Wit et al., 2015; Cullens et al., 2015).

(2) There are teleconnections between high latitudes and the tropics. For example, it is known that the frequency of SSWs is influenced by the direction of the stratospheric quasi-biennial oscillation (QBO) of the zonal wind in the tropics (e.g., Holton and Tan, 1980): SSWs are more frequently observed if the zonal wind in the tropics at 50 hPa is westward. However, there are also indications for influences in the opposite direction: SSWs can influence the tropospheric temperature and convection
in the tropics (e.g., Kodera, 2006), and thereby also tropical clouds and moisture (Eguchi and Kodera, 2010). In particular, SSWs have strong influence on the composition of the stratosphere by modulating the tropical ascent of trace species into the stratosphere (Tao et al., 2015a, b). These teleconnections between tropics and high latitudes are still not fully captured by GCMs/CCMs (e.g., Scaife et al., 2014).





(3) The forecast skill of weather prediction is related to the occurrence of SSWs (e.g., Sigmond et al., 2013; Domeisen et al., 2015), and this can be utilized for an improvement of weather forecasts.

(4) SSWs have effect on the average temperature of the stratosphere at high northern latitudes. Changes in the frequency of SSWs will therefore result in temperature trends (Angot et al., 2012). In addition, the warming of the lower stratosphere induced by SSWs can lead to temperatures too warm for polar stratospheric clouds to form, which has strong influence on ozone depletion in the Arctic. On the other hand, changes in the frequency of SSWs will also affect the meridional circulation and tracer transport from lower latitudes, possibly resulting in changes in ozone depletion of opposite sign (e.g., Konopka et al., 2007). Particularly the timing of SSWs is of importance, with later SSWs favoring stronger ozone depletion (e.g., Kuttipurath and Nikulin, 2012; von Hobe et al., 2013). The timing of SSWs, however, may change in response to climate change. If SSWs are shifted to later in winter, as expected by Naoe and Shibata (2012), stronger ozone depletion would be expected in the Arctic. Therefore, realistic representation of SSWs and their evolution could be important for obtaining more realistic ozone projections for a changing climate.

(5) It has been found that at high latitudes the circulation anomalies of SSWs have influence on surface weather and climate (e.g., Baldwin and Dunkerton, 2001; Kodera et al., 2016), as well as on sea surface temperature (O'Callaghan et al., 2014). A recent review on the mechanism of downward coupling during SSWs is given, for example, by Kidston et al. (2015). Because of this downward influence, representation of SSWs is required for accurate simulation of Northern Hemisphere climate both on global and regional scales (e.g., Gerber et al., 2012, and references therein).

All these effects and interlinks show the importance of understanding all processes that are relevant for the formation and evolution of SSWs, and of including them in GCMs/CCMs. As mentioned before, one important process for the onset and evolution of SSWs is driving by atmospheric waves, particularly by planetary waves. The effect of gravity waves is still not well known. However, there are indications that they significantly contribute.

## 1.2 Effects of gravity waves during SSWs

The effects of gravity waves during SSWs are manifold. For example, in model simulations, the frequency of SSWs does not depend on planetary waves alone, but also on parameterized gravity wave drag (e.g., Richter et al., 2010). Further, the effect of SSWs is not limited to the stratosphere. Also the mesosphere (e.g., Labitzke, 1972; Jacobi et al., 2003; Siskind et al., 2005, 2010; Hoffmann et al., 2007; Yamashita et al., 2013; Zülicke and Becker, 2013), and even the thermosphere/ionosphere are affected (e.g., Goncharenko and Zhang, 2008; Fuller-Rowell et al., 2010; Yigit et al., 2014). The selective filtering of gravity waves by the anomalous winds during (major) SSWs is an important mechanism in these vertical influences. SSWs are associated with mesospheric coolings (e.g., Labitzke, 1972; Siskind et al., 2005; Hoffmann et al., 2007) that are likely driven by dissipation of eastward propagating gravity waves (e.g., Holton, 1983; Miller et al., 2013). As a consequence, the zonal wind in the mesosphere/lower thermosphere (MLT), that is usually directed westward during winter, can change its sign to eastward (e.g., Holton, 1983). A recent review about coupling between stratosphere and mesosphere during SSWs is given by Chandran et al. (2014).





One particular subset of SSWs are polar-night jet oscillation events (PJO events) (e.g., Hitchcock and Shepherd, 2013). These PJO events often are related to strong major SSWs, but sometimes they can be related also to minor warmings. Polar-night jet oscillation events can be characterized as follows: After the peak of the warming, the stratopause altitude drops rapidly within a few days, and the stratopause disappears (e.g., France et al., 2012). This rapid drop of the stratopause is driven by

downwelling induced by breaking planetary waves. Then, after a short period of nearly isothermal conditions in the whole middle atmosphere, a new stratopause forms at altitudes around 75 km, propagates gradually downward with time, and reaches its nominal (climatological) altitude of around 50 km after 1–2 months. Similarly, an eastward directed polar jet re-establishes at elevated altitudes and propagates downward together with the elevated stratopause (e.g., Manney et al., 2008, 2009a, b; Orsolini et al., 2010).

Likely both planetary waves (e.g., Limpasuvan et al., 2012) and gravity waves contribute to the formation of the new elevated stratopause. For example, model simulations show that the formation of the new stratopause is sensitive to nonorographic gravity wave drag (e.g., Chandran et al., 2011; Ren et al., 2011). Further, model simulations that explicitly resolve a considerable part of the gravity wave spectrum reveal that during PJO events the dissipation altitude of westward propagating gravity waves is higher after the SSW than before. Consequently, the residual circulation induced by those gravity waves that is responsible

for the formation of the stratopause is also raised, which explains the formation of the new stratopause at an elevated level (Tomikawa et al., 2012).

During PJO events, gravity waves are also important for the recovery of the eastward directed polar jet after a period of anomalous westward winds. In particular, it has been suggested in a modeling study by Tomikawa et al. (2012) that in cases when a band of anomalous westward winds (induced by the SSW) is located below the eastward jet, westward propagating

gravity waves are filtered out more effectively, and less westward momentum of gravity waves is available to slow down the eastward jet above, and it can therefore reach considerable strength. During this recovery phase of polar jet and stratopause, both the evolution of stratopause height and the descent of tracers from the mesosphere are sensitive to the settings of the gravity wave drag schemes used in model simulations (e.g., Ren et al., 2011), and the importance of both orographic and nonorographic gravity wave drag during the recovery of the polar jet has been pointed out (e.g., McLandress et al., 2013;

Hitchcock and Shepherd, 2013).

Gravity waves could also play an important role for the onset and triggering of SSWs. As has been pointed out by Albers and Birner (2014), before the onset of a SSW, gravity wave drag in the stratosphere is non-negligible. Therefore, gravity waves may contribute to the preconditioning of the polar vortex and its shape such that resonant amplification of planetary wave amplitudes occurs and a SSW takes place. For details see Albers and Birner (2014) and references therein.

Still, the role of gravity waves in SSWs is not fully understood, as has been shown in several recent studies (e.g., Limpasuvan et al., 2011, 2012; Yamashita et al., 2010a, b, 2013). One of the main reasons is that gravity waves have short horizontal scales (tens of kilometers to around thousand kilometers at high latitudes). Therefore their scales are too small to be resolved in most GCMs/CCMs, and their effect on the background flow has to be parameterized. These parameterization schemes are very simplified, and they contain a number of tunable parameters. The improvement of those parameterization schemes by

comparison with global observations is a still ongoing issue (e.g., Ern et al., 2006; Alexander et al., 2010; Geller et al., 2013).





However, also the part of the gravity wave spectrum that is resolved by high resolution models is not necessarily fully realistic, and needs to be validated (e.g., Schroeder et al., 2009; Preusse et al., 2014).

Given the sensitivity of simulated SSWs to uncertain representation of gravity waves in global models, observations of gravity waves (both ground and satellite based) provide a vital tool for the clarification of the role of gravity waves in the evolution of observed SSWs and to evaluate the adequacy of their representation in models.

### 1.3 Gravity wave observations during SSWs

Much work about gravity waves during SSWs has been done using ground-based observations, for example lidar data (e.g., Duck et al., 1998) and radar data (e.g., Hoffmann et al., 2002, 2007). One of the main findings is that during SSWs gravity wave activity in the upper stratosphere and in the mesosphere is related to the background winds, and selective filtering of gravity waves by the background winds is an important effect (e.g., Thurairajah et al., 2010; Matthias et al., 2012). Recent work has derived gravity wave momentum fluxes from radar observations in the upper mesosphere (e.g., de Wit et al., 2014; Placke et al., 2015), and the gravity wave drag on the background flow was estimated during the major SSW in 2013 (de Wit et al., 2014).

Although ground based stations have already provided a wealth of information during SSWs, these observations do not provide a global or zonal average view. In particular, gravity wave activity strongly depends on the exact location of the vortex edge. The shape of the polar vortex, however, will vary considerably during a SSW due to changes in planetary wave amplitudes and phases.

In recent years, also considerable work has been done utilizing satellite observations of gravity waves during SSWs. For example, it has been shown by Wang and Alexander (2009) that sometimes gravity wave amplitudes are positively correlated with warming peaks of SSWs and that the selective filtering by the background wind is important for the propagation of gravity waves into the stratosphere. Also the study by Jia et al. (2015) indicates that the background wind has strong influence on gravity wave activity in the stratosphere. Other studies revealed that sometimes after major SSWs, gravity wave activity in the stratosphere is strongly suppressed (e.g., Wright et al., 2010; Thurairajah et al., 2014). On the other hand, before the major SSW in winter 2005/2006, an enhancement of gravity wave momentum flux was observed in the lower mesosphere (France et al., 2012). Gravity waves may also contribute to the observed tracer descent from the mesosphere after (major) SSWs (Thurairajah et al., 2014). Further, the study by Yamashita et al. (2013) suggests that meridional propagation of gravity waves could be important during major warmings.

Most of these studies are based on either gravity wave amplitudes, variances or potential energies (Wang and Alexander, 2009; Yamashita et al., 2013; Thurairajah et al., 2014; Jia et al., 2015). Studies based on gravity wave momentum fluxes, on the other hand, allow a more direct comparison with gravity wave parameterization schemes. However, so far only few studies of SSWs based on satellite observations used gravity wave momentum fluxes (Wright et al., 2010; France et al., 2012), and these studies are focused mainly on the stratosphere. Momentum flux observations at higher altitudes, as well as direct estimation of gravity wave dissipation from momentum flux vertical gradients (gravity wave potential drag) derived from satellite observations have not been exploited in SSW studies. In addition, previous work using gravity wave momentum flux





observations from space was focused only on single major SSW events, and a more comprehensive comparison of different Arctic winters and different major SSW events is still an open issue.

In our work, we derive absolute gravity wave momentum fluxes and gravity wave potential drag from Sounding of the Atmosphere using Broadband Emission Radiometry (SABER) satellite observations for the Arctic winters 2001/2002 until 2013/2014, and from High Resolution Dynamics Limb Sounder (HIRDLS) satellite observations for the winters 2004/2005 until 2007/2008. This gravity wave activity is compared to atmospheric background conditions provided by SABER, the Microwave Limb Sounder (MLS) on the Aura satellite, and the ERA-Interim reanalysis of the European Centre for Medium-Range Weather Forecasts (ECMWF).

The manuscript is organized as follows: In Sect. 2 the different data sets used in our study are briefly introduced. For the boreal winters 2001/2002 until 2013/2014 the zonal average temporal evolution of atmospheric background temperature and zonal winds averaged over 60° N–80° N is discussed in Sect. 3. In Sect. 4, the altitude-time distribution of gravity wave squared amplitudes, momentum fluxes and gravity wave potential drag, averaged over the latitude band 60° N–80° N, is studied for all considered Arctic winters. In addition, in Sects. 5 and 6 the horizontal distribution of gravity waves, as well as zonal average cross sections of gravity wave squared amplitudes, momentum flux and gravity wave potential drag are investigated before, during, and after the major SSWs of the years 2009 (Sect. 5) and 2006 (Sect. 6). Based on the zonal average cross sections particularly the role of meridional propagation of gravity waves is discussed. Finally, Sect. 7 summarizes our main findings.

## 2 Data sets

Our work is based on data of the satellite instruments MLS, SABER, and HIRDLS, as well as on the ERA-Interim reanalysis. In the following, some information on ERA-Interim and the different satellite instruments is given. Further, we describe how atmospheric background fields are obtained, and how gravity wave amplitudes, momentum fluxes and potential drag are derived from HIRDLS and SABER temperatures.

### 2.1 ERA–Interim

For parts of our study we use meteorological fields (temperature and winds) of the ECMWF ERA–Interim reanalysis (e.g., Dee et al., 2011). Our ERA-Interim data are interpolated on a longitude/latitude grid of 1°×1° resolution. The altitude resolution is about 1.4 km. Global fields are available at 00:00, 06:00, 12:00, and 18:00 UT. Because numerous observations are assimilated in ERA-Interim (Dee et al., 2011), ERA-Interim meteorological fields are considered to be quite reliable in the troposphere and lower and middle stratosphere. At higher altitudes, however, due to a lack of observations, reanalyses become more and more unreliable, particularly during the complicated dynamical situation of SSWs (e.g., Manney et al., 2008). It should also be noted that ERA-Interim simulates the effect of nonorographic gravity waves by Rayleigh friction. A parameterization of nonorographic gravity waves was only introduced in later ECMWF model versions in order to provide more realistic results particularly at higher altitudes (Orr et al., 2010).





## 2.2 The MLS Aura instrument

The Microwave Limb Sounder (MLS) instrument on NASA's Aura satellite is a microwave radiometer that observes atmospheric microwave emissions using the limb sounding method (e.g., Waters et al., 2006; Livesey et al., 2013). MLS observes atmospheric temperature and numerous trace species. In our study, we use MLS version 3.3 temperatures, as well as geopo-
tential height. Useful altitude range is between the 316 and 0.001 hPa pressure levels (between about the middle troposphere and somewhat above the mesopause). Vertical resolution is about 4 km in the stratosphere, and about 14 km in the mesopause region. For more details on the temperature/pressure retrieval and the altitude resolution defined by the averaging kernels see Schwartz et al. (2008). Latitude coverage is between 82° S and 82° N. Measurements are available starting from 08 August 2004 and are still ongoing at the time of writing.

## 10  2.3 The HIRDLS instrument

Like MLS, the HIRDLS instrument was launched on NASA's Aura satellite. From January 2005 until March 2008 HIRDLS observed atmospheric infrared radiances in limb-viewing geometry. From these infrared emissions in limb-viewing geometry, temperature-pressure profiles were derived at altitudes between the tropopause region and well above 70 km. Latitudes between about 63° S and 80° N are covered by HIRDLS observations. In our study we use version V006 HIRDLS temperatures (see
also Gille et al., 2011). Detailed information about the HIRDLS temperature retrieval can be found in Gille et al. (2008). The vertical resolution of observed temperature profiles is about 1 km (e.g., Barnett et al., 2008; Gille et al., 2008; Wright et al., 2011). Along-track sampling distance is about 90 km. Dense along-track sampling, together with good altitude resolution, allows to estimate absolute gravity wave momentum fluxes from observed HIRDLS temperature fluctuations (e.g., Alexander et al., 2008; Wright et al., 2010; Ern et al., 2011).

## 20  2.4 The SABER instrument

The SABER instrument was launched on the Thermosphere-Ionosphere-Mesosphere Energetics and Dynamics (TIMED) satellite. Like for HIRDLS, atmospheric temperatures are derived from atmospheric infrared emissions observed in limb-viewing geometry. SABER temperatures are available from the tropopause region to well above 100 km in the lower thermosphere. SABER switches between southward viewing and northward viewing geometries every about 60 days for about 60 days. The
latitude coverage is either about 82° S–50° N (southward view) or about 50° S–82° N (northward view). This is particularly relevant for our study, because SABER does not continuously observe high northern latitudes. SABER temperature profiles have an altitude resolution of about 2 km, and every second pair of consecutive altitude profiles has an along track sampling of better than 300 km, allowing to derive gravity wave absolute momentum fluxes (e.g., Ern et al., 2011; Ern et al., 2013). In our study, we use SABER version v2.00 temperatures. More information about the SABER instrument is available, for example,
in Mlynczak (1997), or in Russell et al. (1999). Details about the SABER temperature retrieval can be found, for example, in Remsberg et al. (2004, 2008).





## 2.5 Satellite observations of gravity wave amplitudes, momentum fluxes, and potential drag

The determination of gravity wave absolute momentum fluxes from satellite data is a procedure that requires several steps. First, the zonal average background is removed from every altitude profile of observed temperatures. Further, global-scale waves with zonal wavenumbers 1–6 are removed from each altitude profile by reconstructing the contribution of global-scale waves at the exact location and time of each observation. For this purpose, longitude-time spectra are estimated for a fixed set of altitudes and latitudes. This procedure is described in detail in Ern et al. (2011), and it accounts also for fast traveling planetary waves, such as short-period Kelvin waves (e.g., Ern et al., 2008; Ern and Preusse, 2009; Ern et al., 2009) or quasi two-day waves (e.g., Ern et al., 2013). Tides, however, are removed separately, as described in Ern et al. (2013). The result of this procedure are altitude profiles of temperature perturbations that can be attributed to small-scale gravity waves.

For each altitude profile of temperature perturbations the dominant wave structures are identified by a two-step procedure called Maximum Entropy Method/Harmonic Analysis (MEM/HA), as described in detail by Preusse et al. (2002). For these waves, vertical wavelengths and amplitudes are estimated in sliding $10\,\mathrm{km}$ vertical windows, and altitude profiles of vertical wavelengths and amplitudes are obtained.

HIRDLS and SABER are limb sounders that observe the atmosphere with only a single viewing direction. At a given altitude, the satellite soundings are arranged in a measurement track in parallel to the ground track of the satellite. We assume that the same gravity wave is observed in two consecutive altitude profiles of a satellite measurement track, if for a given altitude the vertical wavelengths in these two profiles do not differ by more than 40%. If these pairs of altitude profiles are observed at almost the same location (horizontal separation of less than $300\,\mathrm{km}$) and at the same time (within one minute or less), the projection of the horizontal wavelength of the observed gravity wave on the satellite measurement track can be estimated from the vertical phase shift of the wave structures seen in both profiles (Ern et al., 2004). This estimate of the horizontal wavelength, however, is, in most cases, an upper estimate of the "true" horizontal wavelength of the gravity wave (Preusse et al., 2009a).

Absolute values of gravity wave momentum fluxes $F_{ph}$ can be calculated in the following way (Ern et al., 2004):

$$F_{ph} = \frac{1}{2}\,\varrho_0\,\frac{\lambda_z}{\lambda_h}\left(\frac{g}{N}\right)^2\left(\frac{\hat{T}}{T}\right)^2 \tag{1}$$

In this equation, $\varrho_0$ and $T$ are the atmospheric background density and temperature, $g$ is the gravity acceleration, $N$ the buoyancy frequency, $\lambda_h$ and $\lambda_z$ are the horizontal and the vertical wavelength, respectively, and $\hat{T}$ the temperature amplitude of the wave.

For a limb-viewing instrument with only a single viewing direction, such as HIRDLS or SABER, the uncertainty of these momentum fluxes is large, at least a factor of two. Two of the main shortcomings are biases in the determination of the gravity wave horizontal wavelength, as well as the sensitivity of the instrument to observe gravity waves of a given horizontal and vertical wavelength (observational filter). For a detailed error discussion see Ern et al. (2004). In particular, for instruments with only a single measurement track the direction of momentum flux is not known.

Estimation of momentum fluxes in this way is possible for about 3000 altitude profile pairs per day for HIRDLS, and for about 350 altitude profile pairs per day for SABER. Of course, satellite instruments are sensitive only to part of the whole





spectrum of gravity waves (see also Alexander et al., 2010). In our study, we cover horizontal wavelengths longer than about 100–200 km (e.g., Ern and Preusse, 2012), and vertical wavelengths in the range 2–25 km for HIRDLS, and 4–25 km for SABER (see also Ern et al., 2011). For a detailed discussion of the observational filter of limb observations from satellite see Trinh et al. (2015).

Absolute (total) values of gravity wave drag $XY$ can be estimated from vertical gradients of absolute momentum fluxes:

$$XY = -\frac{1}{\varrho_0}\frac{\partial F_{ph}}{\partial z} \tag{2}$$

As is the case with absolute momentum fluxes, the direction of this drag is not known without additional information. For this reason, we call these values gravity wave "potential drag". Like for absolute momentum fluxes, uncertainties of potential drag are at least a factor of two. Net gravity wave drag could be even zero, while gravity wave potential drag is non-zero, and, of

course, net vectors of momentum flux would be needed in order to estimate the net drag of gravity waves on the background flow.

In situations when the gravity wave spectrum is filtered by strong background winds, it can be assumed that the gravity wave momentum flux spectrum is dominated by waves propagating opposite to the background wind (e.g., Warner et al., 2005). Therefore it can be assumed that at the top of strong wind jets gravity wave potential drag can be used as a proxy for net

gravity wave drag, and the direction of the drag is opposite to the vertical gradient of the wind at the top of the jet. In spite of the large uncertainty of gravity wave potential drag, relative variations of this drag have already led to meaningful results in several cases: for the mesospheric zonal wind jets in the summer hemisphere (Ern et al., 2013), and for both the QBO (Ern et al., 2014) and the semiannual oscillation (SAO) (Ern et al., 2015) of the zonal wind in the tropics. Therefore, it can be expected that meaningful results can also be obtained in our current study for the polar jets in boreal winter.

**3    Atmospheric background conditions in the boreal winters 2001/2002–2013/2014**

In order to characterize the zonal average meteorological background conditions during SSWs, temperatures and winds in both stratosphere and mesosphere are needed. In this section we focus on the latitude range 60° N–80° N. In the stratosphere, temperatures and winds of ERA-Interim have shown to be quite reliable. In addition, SABER and MLS provide temperatures in the whole stratosphere and mesosphere. Further, we derive quasi-geostrophic winds from SABER and MLS pressure and

geopotential fields using the method by Oberheide et al. (2002) that has been applied to SABER data before (Ern et al., 2013).

Because MLS data are not available before August 2004, and SABER observes high northern latitudes for only about 60 days in the months December until March, we compose merged wind fields covering the whole altitude range 20–90 km. Below 40 km we utilize winds from ERA-Interim, at altitudes between 40 and 60 km winds are relaxed with a linear transition toward the SABER or MLS winds, where available, and at altitudes above 60 km only winds from MLS or SABER are used. Because

of their better altitude resolution, we use SABER geostrophic winds, where available, otherwise MLS geostrophic winds are used. Differences between SABER and MLS usually are small. ERA-Interim zonal winds are daily averages, while MLS and SABER geostrophic winds are 3-day averages calculated with a time step of three days and interpolated to obtain daily values.





Altitude-time cross-sections of the merged zonal average zonal winds in $\mathrm{m\,s^{-1}}$ averaged over the latitude band 60° N–80° N, are given in Fig. 1 for the different boreal winters covering the period 2001/2002 to 2013/2014. Temporal coverage is always from 1 December until 31 March. Also given on the x-axis is a time scale in "days of the year" (doy) with 1 January 00:00 UT as doy=0. Areas where no data are available are indicated in gray. Zonal wind contour lines overplotted in Fig. 1 are also given

in all other figures showing altitude-time cross sections.

Figure 2 shows altitude-time cross sections of daily zonal average temperatures in the stratosphere and mesosphere for the boreal winters 2001/2002 until 2013/2014. Figures 2d–2m show temperatures observed by MLS. Different from this, because MLS was not yet in orbit in the first three winters considered, Figs. 2a–2c show temperatures observed by SABER, where available, and ERA-Interim temperatures otherwise (but only at altitudes below 60 km).

In the time period considered, a number of major SSWs took place. These major SSW events can be identified in Fig. 2 by sudden enhancements of the temperatures in the stratosphere. The "central date" of these major SSWs is the day when the polar jet reverses from eastward to anomalously westward wind at 10 hPa (about 32 km altitude) and 60° N. A compilation of major SSW central dates is given, for example, by Charlton and Polvani (2007) for the years 1958 until 2002, or by Cohen and Jones (2011) for the years 1958 until 2010. A compilation of recent central dates is listed in Table 1 for the time period considered in

our study. This compilation is mainly based on the references Cohen and Jones (2011) and Chandran et al. (2013). In addition, the type of the major SSW is classified by either "D" for "displacement" or "S" for "split", depending on whether the polar vortex was either displaced by a strong planetary wave number one, or split into two parts by a strong planetary wave number two (see also Charlton and Polvani, 2007). Further, it is noted by "ES" if an elevated new stratopause forms after the warming. A classification of SSWs can also be found in Chandran et al. (2014), including their type. Of course, the classification of

SSWs into major and minor SSWs somewhat depends on the data, as well as on the criterion used (e.g., Charlton and Polvani, 2007; Butler et al., 2015).

Major SSWs that were associated with a PJO event took place in the winters 2003/2004, 2005/2006, 2008/2009, 2009/2010, and 2012/2013. During these warmings, the stratopause altitude drops, and a new elevated stratopause forms around 75 km altitude. After the central date a new eastward directed polar jet re-establishes at higher altitude and gradually propagates

downward in altitude, while the anomalous westward winds associated with the major SSW persist for a longer time (for one month and longer) in an altitude range below the newly formed eastward polar jet.

In the winter 2011/2012 another PJO event took place. However, because the winds only reversed above the 10 hPa level at 60° N the criterion for a major warming is not matched (see, for example, Chandran et al., 2014), and the SSW in January 2012 is classified only as a minor SSW, even though after the warming a new elevated stratopause is formed in the mesosphere

(Chandran et al., 2013). If an average over a wider latitude range of 60° N–80° N is considered, like in Fig. 2k, the wind reversal in the winter 2011/2012 reaches as low as about 30 km. Therefore, averaged over this latitude band, the response of gravity waves to this change in the global circulation may be similar as for a major warming.

There are also several major SSWs that were not associated with a PJO event, for example in the winters 2001/2002, 2002/2003, and in late winter 2006/2007. In addition, there was a pronounced oscillation of the polar jet during the winter

2007/2008: three minor warmings with anomalous westward winds only in the upper stratosphere were followed by a weak





major warming on 22 February. The only winters that were only little perturbed and had no major SSW or PJO event are 2004/2005, 2010/2011, and 2013/2014.

## 4 Time series of gravity wave activity and gravity wave potential drag

In this section, the zonal average gravity wave activity (gravity wave squared amplitudes and absolute momentum fluxes), as well as gravity wave potential drag derived from vertical gradients of absolute momentum fluxes are investigated. Figure 3 shows gravity wave squared amplitudes in the latitude band $60°$ N–$80°$ N for the winters 2001/2002–2013/2014. Values are given in $\mathrm{K}^2$ on a logarithmic scale. For the same years, Fig. 4 shows absolute gravity wave momentum fluxes in mPa on a logarithmic scale, and Fig. 5 gravity wave potential drag in $\mathrm{m\,s^{-1}\,day^{-1}}$, also on a logarithmic scale. Please note that, as detailed below, gravity wave parameters are only available from HIRDLS and SABER observations in limited time and altitude ranges. Therefore, compared to Figs. 1 and 2, in Figs. 3–5 there are much larger areas indicated in gray where no gravity wave data are available.

In the winters 2001/2002 until 2004/2005 and the winters 2008/2009 until 2013/2014 only SABER gravity wave observations are available. SABER gravity wave data are most reliable at altitudes above $30\,\mathrm{km}$. Therefore in Figs. 3–5 SABER data are shown only above $30\,\mathrm{km}$. In the winters 2005/2006, 2006/2007, and 2007/2008, the values in Figs. 3–5 are a combination of HIRDLS observations at altitudes below $55\,\mathrm{km}$, and of SABER observations at altitudes above. HIRDLS observations are daily averages, while SABER observations are 3–day averages calculated with a time step of one day. During time periods when observations from both instruments are available, in the altitude range $50$–$55\,\mathrm{km}$ a smooth transition is made between both data sets. Overplotted contour lines in Figs. 3–5 indicate the zonal average zonal wind, determined as described in Sect. 3. Contour interval is $20\,\mathrm{m\,s^{-1}}$, eastward (westward) wind is indicated by solid (dashed) contour lines, the zero wind line is indicated in bold solid.

Atmospheric background winds have strong influence on the global distribution of gravity wave activity. Because zonal winds usually are much stronger than meridional winds, their influence on the gravity wave distribution is also much stronger, and we will focus on the effect of the zonal average zonal wind $\overline{u}$ on zonal average gravity wave distributions.

There are two main processes related to the background wind that shape the gravity wave distribution. The first process is critical level filtering: During wave propagation it can happen that background wind $\overline{u}$ and ground based phase speed $c_\varphi$ of a gravity wave become equal. In this case, the intrinsic phase speed of the wave $\widehat{c}_\varphi = c_\varphi - \overline{u}$ becomes zero, the wave cannot propagate further and it dissipates completely.

The second process is wave saturation. If a gravity wave propagates conservatively upward (without dissipation), the wave amplitude will grow exponentially according to the decrease in background density. At some point, the wave amplitude cannot grow further. The wave amplitude reaches its saturation limit, and the wave breaks. This can happen without a critical level being reached. The saturation temperature amplitude ($\widehat{T}_{sat}$) is proportional to the intrinsic phase speed, i.e., to the difference between ground based phase speed and background wind:

$$\widehat{T}_{sat} = \frac{T}{g} N |c_\varphi - \overline{u}| \tag{3}$$





The saturation momentum flux of a gravity wave is proportional to the intrinsic phase speed to the power of 3. See also Preusse et al. (2006), Eq. (10) in Ern et al. (2008), and the discussion in Ern et al. (2015).

## 4.1 Discussion of gravity wave squared amplitudes

In the upper mesosphere gravity wave squared amplitudes are usually quite high ($30\,K^2$ and higher), and there is little interan-
nual variation. Main differences are found in the stratosphere and in the lower mesosphere. These differences will be discussed in the following for different vortex conditions.

### 4.1.1 Strong polar jets

Situations of strong unperturbed (i.e., continuously eastward directed) stratospheric polar jets are found, for example, in the winters 2004/2005, 2010/2011, and during December 2006 until mid February 2007, during December 2007 until mid January
2008, as well as during the first half of January 2013/2014.

During these unperturbed periods, on zonal average there is no wind reversal in the stratosphere and lower mesosphere. Under these conditions, gravity waves with westward or zero ground based phase speeds can propagate in this whole altitude range without encountering critical levels. This means that those gravity waves can attain large amplitudes already in the stratosphere and lower mesosphere, because their intrinsic phase speeds and thus their saturation temperature amplitudes are
high. This effect is clearly seen in Fig. 3: during the mentioned periods of strong unperturbed polar jets gravity wave squared amplitudes are quite high with values of about $5\,K^2$ around $30\,km$ altitude, and of about $20\,K^2$ around $50\,km$ altitude.

### 4.1.2 Weak polar jets

Compared to the situation of strong polar vortices, during weak vortex conditions (zonal mean zonal wind weaker than about $20\,m\,s^{-1}$), gravity wave squared amplitudes in the mid stratosphere around $30\,km$ altitude are somewhat reduced (about $3\,K^2$).
These conditions are found, for example, during parts of the winter 2001/2002, during much of the winter 2002/2003 (Fig. 3b), and during winter 2013/2014 in the second half of January and during February. Likely reason for this reduced gravity wave activity are reduced gravity wave saturation amplitudes that are not enhanced by strong favorable background winds.

### 4.1.3 PJO events

In Fig. 3, the highest values of gravity wave squared amplitudes (about $10\,K^2$ and more) in the mid stratosphere (at $\sim 30\,km$
altitude) are found around the central date of strong major SSWs with PJO event, i.e., when the amplitude of planetary waves maximizes. This is the case, for example, for 2008/2009 around January 24 (Fig. 3h), or for 2012/2013 around January 7 (Fig. 3l). For the 2009/2010 PJO event, gravity wave squared amplitudes are enhanced around January 25, somewhat before the central date February 9 (Fig. 3i). Also for the minor SSW in January 2012 that is associated with a PJO event (Fig. 3k) gravity wave squared amplitudes in the mid stratosphere are somewhat enhanced during the period of anomalous westward
winds in mid January. On the other hand, around the central date January 21 of the 2005/2006 PJO event (Fig. 3e), gravity





wave squared amplitudes are not so much enhanced. Since zonal average wind speeds are not necessarily stronger than during other situations, enhancements of squared amplitudes close to the SSW central dates are probably not only caused by favorable propagation conditions, but also by stronger activity of gravity wave sources.

After the onset of SSWs associated with PJO events (not multiple SSWs associated with a single PJO), gravity wave activity in the stratosphere is reduced for two reasons. First, zonal winds are much weaker, not resulting in favorable enhancements of gravity wave saturation amplitudes. Second, due to anomalous westward winds there are wind reversals in the troposphere and/or in the stratosphere, such that gravity waves with zero ground based phase speed (e.g., mountain waves) or with slow westward directed phase speeds will encounter critical levels. Particularly during the phases of PJO events when weak anomalous westward winds persist for 1–2 months below the newly forming eastward jet, i.e. well after the SSW central date, gravity wave squared amplitudes are quite low in the whole stratosphere (as low as $1$–$2\,\mathrm{K}^2$). This is the case for all PJO events in the time period considered — even for the PJO event in winter 2011/2012 that is associated with only a minor SSW, although the zonal wind in the mid and lower stratosphere only weakens after the SSW, and does not reverse on zonal average in the latitude range considered.

In the six PJO events considered, a very strong eastward polar jet re-establishes in the upper mesosphere and slowly descends into the stratosphere within 1–2 months. In the lower part of this re-established polar jet, gravity waves over a wide range of eastward phase speeds will encounter critical levels, dissipate, and it could be expected that they might significantly contribute to the formation and maintenance of this jet. However, as will be shown later in Sect. 4.3, this is likely not the case.

High westward phase speed gravity waves are not filtered out by the comparably weak anomalous westward winds in the troposphere and lower stratosphere. In the strong eastward jets, their critical amplitudes can become very large (cf. Eq. 3). This is one possible explanation why very high gravity wave squared amplitudes of well above $30\,\mathrm{K}^2$, i.e. values similar to those in the other years, are found in the upper part of the re-established eastward jets and above (at altitudes above about 70 to $80\,\mathrm{km}$). Another explanation for these high squared amplitudes could be meridional propagation of gravity waves from lower latitudes, as suggested by Yamashita et al. (2013).

### 4.1.4 Other SSWs

After other SSWs, the zonal wind in the stratosphere is usually much weaker than before. Consequently, there is no favorable enhancement of gravity wave saturation amplitudes like in the strong polar jets of unperturbed winters, or like often before SSWs. In addition, during periods of anomalous westward winds gravity waves with zero or slow westward directed phase speeds will encounter critical levels. The result are reduced gravity wave squared amplitudes in the stratosphere and sometimes the lower mesosphere. This is seen, for example, in winter 2001/2002 after mid February, in late winter / early spring 2006/2007 after mid February, and after the major SSW in 2008 after mid February.

Around the central dates of other major SSWs or just before minor SSWs, in the latitude range considered, there are no clear enhancements of gravity wave squared amplitudes in the mid stratosphere. However, there are four pulses of enhanced gravity wave variances in the lower stratosphere (below $30\,\mathrm{km}$ altitude) in January and February 2008 (Fig. 3g) that are related



to enhanced planetary wave amplitudes during three minor SSWs and one major SSW. This effect has been discussed in detail by Wang and Alexander (2009) for the winter 2007/2008.

Together with the findings from the PJO events, this shows that there is no clear relationship between onset of stratospheric warming (central date) and strength of gravity wave squared amplitudes. This indicates that the shape and location of the polar vortex is important in determining the strength of observed gravity wave activity: Depending on the shape of the vortex, enhancements of jet-related gravity wave source processes could be expected. In particular, the strongest enhancement of gravity wave squared amplitudes is found for the central dates of the 2008/2009 and 2012/2013 major SSWs, which were both vortex split events. This is qualitatively in good agreement with the findings by Albers and Birner (2014) who found enhanced orographic gravity wave drag for split vortex events in the ERA-Interim reanalysis and in the Japanese Meteorological Agency and Central Research Institute of Electrical Power Industry 25-year Reanalysis (JRA-25). One possible reason for enhanced gravity wave activity during vortex-split events could be the strong jet curvature and the existence of two jet exit regions that will lead to enhanced jet-related gravity wave sources. But also the location of the polar vortex should be important. For example, stronger gravity wave activity would be expected if the polar jet crosses mountain ranges, resulting in stronger excitation of mountain waves. For this reason, the importance of the particular conditions of the polar vortex will be discussed later in more detail in Sects. 5 and 6 for the 2008/2009 vortex split event and for the 2005/2006 vortex displacement event.

## 4.2 Discussion of gravity wave momentum fluxes

Much of the discussion of gravity wave squared amplitudes in Sect. 4.1 is also valid for gravity wave momentum fluxes and will therefore not be repeated in detail. The main difference is that gravity wave amplitudes usually grow with altitude. If a gravity wave propagates conservatively, this amplitude growth is exponential, compensating the exponential decrease of atmospheric background density with altitude.

Different from this, gravity wave momentum flux is conserved, i.e. remains constant, if a wave propagates conservatively. In all panels of Fig. 4, however, we find that gravity wave momentum flux gradually decreases with altitude, indicating an overall dissipation of gravity waves while the waves are propagating upward.

There are several further findings. First, in situations of strong polar jets, an increased amount of gravity wave momentum flux enters the stratosphere. Like for gravity wave variances, this is the case during the periods of strong polar jets in the winters 2004/2005, 2010/2011, 2013/2014 (see Figs. 4d, 4k, 4m), and during December 2006 until January 2007 (see Fig. 4f) as well as in December 2007 (see Fig. 4g). Second, for the cases when gravity wave squared amplitudes are enhanced shortly before the central dates of major SSWs, or before the onset of a minor SSW (i.e., mainly before most of the PJO events), momentum fluxes are also enhanced. This is likely caused by enhanced planetary wave activity having effect both on gravity wave propagation conditions and on gravity wave sources (e.g., mountain waves and jet-related source processes). And, third, gravity wave momentum fluxes are reduced in the stratosphere, when zonal winds are weak.

For the PJO events, during the 1–2 month phases of anomalous stratospheric westward winds persisting after the SSW, we find that gravity wave momentum fluxes are strongly reduced in both stratosphere and mesosphere. This is the case particularly when the polar jet starts to re-establish at elevated altitudes. During this period momentum flux vertical gradients are sometimes





close to zero, while negative gradients would be expected. This again might indicate that meridional propagation of gravity waves could play an important role and additional momentum flux is transported from lower latitudes into the latitude band 60° N–80° N.

### 4.3 Discussion of gravity wave potential drag

While gravity wave squared amplitudes and gravity wave momentum fluxes have already indicated that there are strong interactions between gravity waves and the background winds, calculation of gravity wave potential drag can serve as a metric whether the variations seen could have significant effect on the background flow, or not. Considering Fig. 5, there are a number of noteworthy findings:

(1) In the stratosphere, usually there seems to be no particular enhancement of gravity wave potential drag that would
contribute to the onset of SSWs directly in the latitude range considered (i.e. 60° N–80° N).

(2) In the case of strong, unperturbed stratospheric polar jets (see also Sect. 4.1.1) gravity wave potential drag in the range 1–$3\,\mathrm{m\,s^{-1}\,day^{-1}}$ is found in the stratosphere. This suggests that wave driving by gravity waves somewhat contributes to the zonal momentum budget for unperturbed conditions. These values are similar to the drag due to planetary waves during unperturbed conditions. This supports the findings by Albers and Birner (2014) that stratospheric gravity wave drag before SSWs is non-
negligible and gravity waves could therefore be important for preconditioning the polar vortex such that a SSW is triggered. For perturbed conditions, however, the drag due to planetary waves can be up to around $30\,\mathrm{m\,s^{-1}\,day^{-1}}$ in the stratosphere (see, for example, Hitchcock and Shepherd, 2013), i.e. considerably stronger.

(3) When background winds are weak (see also Sect. 4.1.2) only little gravity wave potential drag is found in the stratosphere ($1\,\mathrm{m\,s^{-1}\,day^{-1}}$ and below). The likely reason is that only a reduced amount of gravity wave momentum flux can enter the
stratosphere: due to the weak background winds, there is no favorable enhancement of gravity wave saturation amplitudes.

(4) Momentum fluxes are even more reduced in the case of wind reversals in the troposphere and lower stratosphere, as is the case during PJO events when anomalous westward winds are persisting in the stratosphere after the SSW. Consequently, during these periods, gravity wave potential drag in the stratosphere is also much weaker. This is the case for all PJO events: the major SSWs in the winters 2003/2004, 2005/2006, 2008/2009, 2009/2010, and 2012/2013, as well as for the minor warming
in the winter 2011/2012.

(5) In the cases when during a PJO event a strong polar jet re-establishes after the SSW, one might expect that enhanced gravity wave potential drag would be seen in the lower part of the newly formed polar jet, because gravity waves with eastward phase speeds will encounter critical levels for a wide range of ground based phase speeds (in some cases even about 10–$80\,\mathrm{m\,s^{-1}}$). In Figs. 5c, 5e, 5h, 5i, 5k, and 5l, however, only very weak gravity wave potential drag is seen in the lower part of
the re-established eastward jets.

One possible explanation for this finding is that, for these situations, background winds in almost the whole stratosphere are quite weak. Consequently, also gravity wave saturation amplitudes will be quite low in almost the whole stratosphere over a large altitude range. This considerably reduces the amount of gravity wave momentum flux that is available for interacting with the background winds in the lower part of the eastward jets. Another possible explanation could be that gravity wave activity





in boreal winters may be dominated by gravity waves with slow ground based phase speeds, for example mountain waves, as indicated in previous model-measurement comparisons (e.g., Preusse et al., 2009b). On the one hand, these waves would encounter critical levels in the troposphere or lower stratosphere due to the wind reversals caused by anomalous westward winds in the lower stratosphere. And, on the other hand, having only low ground based phase speeds, these waves are too slow

to contribute to the formation of wind jets with zonal wind speeds as high as $80\,\mathrm{m\,s^{-1}}$.

Obviously, the momentum flux of high eastward phase speed gravity waves is too small to produce significant driving of the eastward jets in their lower parts. Consequently, the re-formation of the eastward polar jets after major SSWs is likely an effect induced by an anomalous meridional residual circulation, and not directly driven by gravity waves or planetary waves. This finding is in good agreement with modeling studies. For example, Hitchcock and Shepherd (2013) point out that the situation

of the re-established polar jets during PJO events is very different from wave-driven circulations like the QBO. For the QBO enhanced wave drag is seen both on top and at the bottom of an eastward or westward directed wind jet, i.e. for both positive and negative vertical shear of the zonal wind (see also Ern et al., 2014). For the re-established eastward directed polar jets, however, gravity wave drag is only enhanced at the top of the jet. As suggested by Hitchcock and Shepherd (2013), at the top of the jet wave saturation effects should be more important than critical level filtering.

The absence of strong gravity wave drag on the lower flank of the jet indicates that the new polar jet is caused by stratospheric cooling after the SSW and the response of the residual circulation to this cooling (Tomikawa et al., 2012; Hitchcock and Shepherd, 2013). The importance of thermal driving and the resulting residual circulation is confirmed by several studies that observe enhanced transport of trace species from the mesosphere downward, induced by an enhanced meridional and downward directed residual circulation (e.g., Manney et al., 2009a, b; Orsolini et al., 2010; Salmi et al., 2013; Bailey et al.,

2014). In addition to the observed tracer transport, the long time required until the newly formed stratopause reaches the climatological stratopause altitude (see Sect. 3) indicates that the anomalous residual circulation accompanying a PJO event persists for a longer time after the central date of the SSW.

(6) While gravity wave driving is not observed in the lower part of the re-established polar jets during PJO events, considerable gravity wave potential drag of well above $10\,\mathrm{m\,s^{-1}\,day^{-1}}$ is observed in the upper part of these jets. From meteor

radar observations, even stronger gravity wave drag of about $100\,\mathrm{m\,s^{-1}\,day^{-1}}$ is reported (de Wit et al., 2014). However, these particular measurements are from a location that is known for enhanced activity of mountain waves and may therefore not be representative for zonal averages. Further, the quite high gravity wave momentum fluxes seen by meteor radars are currently under debate (e.g., Riggin et al., 2016).

During PJO events high values of potential drag in the upper mesosphere are observed even though momentum fluxes in

the lower stratosphere are much reduced after the SSW. These high values of potential drag are comparable to those in the upper mesosphere during most other periods considered in our study (see Fig. 5). As already indicated by the quite strong gravity wave amplitudes in the altitude range 70–80 km (see Sect. 4.1), likely a mixture of gravity waves that have propagated from lower latitudes, as well as vertically propagating gravity waves with westward directed ground based phase speeds act to decelerate the re-established polar jets in their upper part.





Overall, this suggests that gravity waves contribute to the wind reversal of the re-established polar jets at their top, and, consequently, to the downward propagation of the newly formed stratopause to its nominal altitude around 50 km. The issue of meridional propagation of gravity waves will be addressed again in Sects. 5 and 6.

(7) From Fig. 5, it can also be seen that for the PJO events around the central date of the major SSWs, or, in the case of the PJO event during winter 2011/2012, around the onset of the minor SSW, gravity wave potential drag in the upper mesosphere usually is not reduced. This might indicate that gravity wave drag could be involved in the formation of the new elevated stratopause around 75 km altitude.

(8) Finally, it should be mentioned that in all winters considered enhanced values of gravity wave potential drag are preferentially found in the upper stratosphere and in the mesosphere (i.e., at altitudes above about 40 km). In many cases, these enhanced values are related to vertical shear of the background wind. This is not only the case for the upper part of eastward directed polar jets, but also for some occasions of positive vertical shear of anomalous westward winds at altitudes above 40 km, for example in the second half of January 2006 (Fig. 5e), in mid January 2012 (Fig. 5k), and in the first half of January 2013 (Fig. 5l).

## 5 Gravity waves during different phases of the major SSW in 2009

In this section we will address the effect of gravity waves during different phases of the major SSW in boreal winter 2008/2009. There are several reasons for choosing this major SSW. First, this SSW is associated with a PJO event. This means that after the SSW anomalous westward winds persist for a longer time in the stratosphere (from about 21 January until end of February), a new elevated stratopause is formed, and a new strong eastward directed polar jet is re-established after the SSW. Second, during this SSW strong activity of the quasi-stationary planetary wave with zonal wavenumber 2 is observed, leading to a split vortex. Third, SABER is observing high northern latitudes already somewhat before the central date of the SSW.

### 5.1 Horizontal distributions during the 2009 major SSW

Figure 6 illustrates the temporal evolution of the polar vortex and of gravity wave activity during the different phases of the 2009 major SSW / PJO event at 30 km altitude. Shown are horizontal distributions of ERA-Interim temperatures (left column), zonal wind (second column), absolute horizontal wind (third column), SABER gravity wave squared amplitudes on a logarithmic scale (fourth column), and SABER gravity wave momentum fluxes on a linear scale (right column). For comparison, the first row shows an average over an unperturbed vortex period during 13–28 February 2011, while the other rows show different phases before, around and after the central date (24 January) of the 2009 major SSW.

### 5.1.1 Unperturbed vortex during 2011

For an unperturbed vortex situation, there is a temperature minimum centered at the pole (Fig. 6a1), and the polar jet is strong and axisymmetric around the pole (Figs. 6a2 and 6a3). Enhanced gravity wave activity is usually found in regions of high wind speed (Figs. 6a4 and 6a5). This enhancement may be a consequence of enhanced saturation amplitudes of gravity waves having





phase speeds opposite to the background wind. Still, there are some regions where momentum fluxes are more enhanced. This could be an indication of localized gravity wave sources, for example jet-related sources or orography. Of course, the period 13–28 February 2011 represents conditions of a very stable and axisymmetric polar vortex. Usually, even for widely unperturbed conditions, there will be some displacement (activity of planetary wave number 1) and/or elongation of the vortex (activity of

planetary wave number 2).

### 5.1.2    Well before the central date of SSW 2009

The second row in Fig. 6 shows horizontal distributions for the period 12–16 January 2009, i.e. well before the central date of the 2009 major SSW. The temperature at the pole is still quite low (Fig. 6b1), but the polar vortex is already somewhat perturbed. As a consequence of activity of planetary wave number 2, it is elongated towards North America and North Asia

(Figs. 6b2 and 6b3). Further, there seem to be two jet exit regions (i.e., regions of strong jet deceleration), one over North America, and another one close to Scandinavia. In the vicinity of those jet exit regions we find strongly enhanced gravity wave momentum fluxes (Fig. 6b5). These gravity waves are likely a mixture of jet-generated waves and orographically induced waves by the Rocky Mountains and the Norwegian Alps. Although the jet exit regions are seen at 30 km altitude, we expect that they are a feature persistent over a larger altitude range, and the sources of the jet-generated gravity waves could therefore

be well below 30 km. A review of jet-related gravity wave source processes is given, for example, by Plougonven and Zhang (2014). Somewhat enhanced momentum fluxes are also found over other mountainous regions, such as Northeast America or the Ural Mountains. For an overview of regions that are known as sources for orographically generated gravity waves see, for example, Jiang et al. (2004a), or Hoffmann et al. (2013).

The period 12–16 January 2009 almost coincides with the period 11–15 January 2009 investigated in Albers and Birner

(2014). The distribution of gravity wave momentum fluxes in our Fig. 6b5 is in remarkable agreement with the distribution of orographic gravity wave drag derived from the JRA-25 and ERA-Interim reanalyses, particularly over North America (see Albers and Birner, 2014, their Figs. 6d and 7d). This is an important finding because Albers and Birner (2014) state that, for the conditions prior to the 2009 major SSW, gravity wave forcing, particularly in the longitude range between 60° W and 160° W, could trigger the evolution from an elongated to a peanut-shaped vortex, finally leading to the SSW and vortex split to happen.

Of course, there are some differences between the distributions of orographic gravity wave drag by Albers and Birner (2014) and the gravity wave momentum fluxes in Fig. 6b5. For example, in Fig. 6b5 there is a strong enhancement of gravity wave momentum fluxes over Europe that is not seen in Albers and Birner (2014) (their Figs. 6d and 7d). Such differences may be due to the fact that the satellite observations contain not only mountain waves, but also gravity waves from jet-related sources that are not covered by the analysis of Albers and Birner (2014).

### 30  5.1.3    Shortly before the central date of SSW 2009

The third row in Fig. 6 coincides with the maximum of gravity wave squared amplitudes and momentum fluxes shortly before the central date of the 2009 SSW (see also Figs. 3h and 4h). During this period (17–21 January), there are two distinct temperature minima off the pole (Fig. 6c1), and the vortex is extremely elongated with the polar jet extending even to the





Gulf of Mexico and far into Central Asia (Figs. 6c2 and 6c3). While one of the jet exit regions is still located above North America, the other one has somewhat shifted toward Asia. Accordingly, we find hot spots of gravity wave momentum fluxes in the vicinity of these jet exit regions. One hot spot is located over central North America, and the other over Central Asia. Again, enhanced momentum fluxes are found over mountainous regions, like Northwest America, the Ural Mountains or Scandinavia.

Compared to the period of 12–16 January, however, the momentum fluxes over Scandinavia are much reduced. One possible reason could be the northward shift of the polar jet, another reason could be the shift of the jet exit region towards Asia. It is also noteworthy that enhanced gravity wave activity is found even at latitudes as low as 30° N. There are also two regions of weak westward directed wind, apparently some outflow of the polar vortex that seems to be a first indication of vortex instability and breaking of the planetary wave number 2. One region is located over the North Pacific, and the other over the

Mediterranean. Similar to the polar jet, these regions of enhanced winds could provide favorable propagation conditions for gravity waves. While no enhancement of gravity wave activity is found in the North Pacific region, indeed, enhanced gravity wave momentum fluxes are found over the Mediterranean. Another enhancement of momentum fluxes over the North Atlantic might also be related to this secondary circulation and the breaking of the planetary wave number 2. This, however, is difficult to decide from the gravity wave observations alone.

**5.1.4    Around the central date of SSW 2009**

The period of 22–26 January, which is centered around the central date of the 2009 SSW (24 January), is shown in the fourth row of Fig. 6. As seen in Fig. 6d1, now there is a pronounced temperature maximum close to the pole, as expected for a SSW. At the same time, the polar vortex has weakened and split into two sub-vortices, and two regions of anomalous westward winds are located close to the pole (Figs. 6d2 and 6d3). Additionally, the two wider regions of westward winds at lower latitudes over

the North Pacific and over the Mediterranean have strengthened. As mentioned before, this may be an outflow of the polar vortex and related to rising vortex instability and breaking of the planetary wave 2.

From Figs. 6d4 and 6d5 we find that gravity wave activity has somewhat weakened, compared to the period directly before the central date. Still, some gravity wave activity will be caused by localized orographic sources. However, gravity wave momentum fluxes are similarly enhanced over the whole area of the two vortices, suggesting that the strong jet curvature leads

to a wide distribution of jet-related gravity-wave-generating processes. Further, there is a large area of enhanced momentum fluxes over the North Pacific that coincides with an area of low latitude anomalous westward winds in this region (see Fig. 6d2), and thus this enhancement may be related to the breaking of the planetary wave 2. At the same time, however, gravity wave momentum flux is not much enhanced in the other region of low latitude anomalous winds over the Mediterranean. This shows that breaking planetary waves can act as gravity wave sources. The strong variation of gravity wave activity during this process,

however, indicates that these source processes may be very intermittent.

**5.1.5    Anomalous winds shortly after the central date of SSW 2009**

The fifth row in Fig. 6 shows average horizontal distributions for the period 25–29 January 2009, i.e. shortly after the central date of the SSW. Temperatures still show a zonal wave number 2 structure with two temperature maxima in the latitude range





40° N–80° N (Fig. 6e1). The two sub-vortices of the vortex split event are still clearly visible, and the horizontal separation between these two vortices has considerably grown (Fig. 6e3). The zonal wind displays a zonal wave number 2 pattern of alternating positive (=eastward) and negative (=westward) winds at all latitudes north of 30° N. In the latitude range 60° N–80° N negative winds are much stronger, such that the zonal wind is negative (anomalously westward) on zonal average. At latitudes 30° N–50° N positive winds are stronger, but the regions of negative winds are more extended resulting in close to zero winds on zonal average (Fig. 6e2).

We still find considerable gravity wave activity related to the polar vortices (Figs. 6e4 and 6e5). Enhancements are found in regions of strong jet curvature (above North America and, much weaker, over central Asia), as well as over Eastern Europe, possibly related to the jet exit region. The source altitude, however, could be well below 30 km. Of course, also mountain waves will play an important role. There is also some remaining gravity wave activity over the North Pacific that seems to be related to the anomalous westward winds in this region. It should also be noted that, due to the off-pole displacement of the two vortices, winds are much reduced north of 70° N. Further, there are two regions of anomalous westward winds north of 60° N, which may increase the probability of mountain waves to encounter wind reversals (i.e. critical levels). Both vortex displacement and anomalous winds may therefore contribute to the overall reduction of gravity wave activity at latitudes north of 60° N (see also Figs. 3h and 4h).

### 5.1.6 Extended phase of stratospheric anomalous winds

The sixth row in Fig. 6 covers the time period 9–23 February 2009, i.e. the extended period of stratospheric anomalous westward winds in the latitude range 60° N–80° N (cf. Fig. 1h). During this period, there is little structure in the temperature distribution of the Northern Hemisphere, with a polar temperature minimum just starting to form (Fig. 6f1). The polar vortices have disappeared, and zonal wind is generally weak. There is an almost axisymmetric band of weak anomalous westward winds at latitudes north of about 50° N, while winds are prevalently eastward south of about 50° N (Figs. 6f2 and 6f3).

Due to the very weak winds in the whole Northern Hemisphere, gravity wave activity is also strongly reduced (Figs. 6f4 and 6f5). Partly, this is the case because there are no favorable enhancements of gravity wave saturation amplitudes by the background winds. And, particularly at latitudes north of 50° N, due to the anomalous westward winds there is an increased probability for mountain waves to encounter critical levels before reaching the altitude level of 30 km displayed in Fig. 6. The highest values of gravity wave activity are found in the vicinity of Japan and the Korean Peninsula (see Fig. 6f4).

### 5.1.7 Shortly after the period of anomalous winds

The bottom row of Fig. 6 represents an average over the period 5–15 March 2009. During this period, zonal average zonal winds in the latitude range 60° N–80° N are no longer anomalous and have turned to eastward again in the stratosphere (cf. Fig. 1h). A new temperature minimum has formed at the pole (Fig. 6g1), and the winds at 30 km altitude have started to strengthen again, particularly at latitudes around 40° N (see Figs. 6g2 and 6g3).

As a consequence, gravity wave activity has started to increase again, particularly south of 50° N. Squared amplitudes and momentum fluxes are maximum over Northeast Asia, even reaching as far north as about 60° N. Compared to January 2009,



however, momentum fluxes are still relatively low, likely because of the still quite weak background winds and thus comparably low gravity wave saturation amplitudes.

## 5.2 Zonal average cross sections during the 2009 major SSW

Next, we will investigate zonal average cross sections of MLS temperatures, SABER gravity wave squared amplitudes, absolute momentum fluxes, and gravity wave potential drag during the major SSW in winter 2008/2009 for the same periods as discussed in Sect. 5.1. The results are given in Fig. 7 for temperatures (upper row), gravity wave squared amplitudes (second row), momentum fluxes (third row), and gravity wave potential drag (bottom row) in the latitude range $20°$ N–$90°$ N. Overplotted contour lines are zonal average MLS geostrophic zonal winds with eastward (westward) winds indicated by solid (dashed) contour lines. Zero zonal wind is indicated by a bold solid contour line. Contour increment is $20\,\mathrm{m\,s}^{-1}$.

### 5.2.1 Unperturbed vortex during 2011

The left column of Fig. 7 shows zonal average temperatures, gravity wave squared amplitudes, momentum fluxes and potential drag for the characteristic situation of an unperturbed polar jet during February 11–26 2011. The temperature structure during this period displays the typical wintertime split stratopause pattern (Fig. 7a1) with the polar temperature enhancement being an effect of adiabatic heating by the downwelling branch of the Brewer Dobson circulation (e.g., Hitchman et al., 1989). The polar jet is tilted equatorward (with increasing alitude), displaying the well-known funnel-like shape of an unperturbed polar vortex.

For unperturbed vortex conditions, stratospheric gravity wave squared amplitudes and momentum fluxes are enhanced only at high latitudes where the strong polar jet is located (Figs. 7b1 and 7c1). At higher altitudes, the momentum flux maximum shifts from around $65°$ N in the stratosphere to about $45°$ N in the mesosphere, which might be an indication for gravity waves propagating from higher to lower latitudes while propagating upward, following the tilted polar jet. We also find enhanced gravity wave potential drag at latitudes $55°$ N–$80°$ N in the altitude range 45–55 km where the polar jet significantly weakens. Obviously, this gravity wave potential drag has a net decelerating effect at the top of the polar jet. Another enhancement of gravity wave potential drag is found in the upper mesosphere with particularly high values around the zero wind line at the top of the polar jet equatorward of about $55°$ N. In the lower part of the polar jet, where zonal wind vertical gradients are positive, gravity wave potential drag is not as strong as in the upper part of the jet.

### 5.2.2 Well before the central date of SSW 2009

In the period 12–16 January 2009, i.e. well before the central date of the major SSW 2009, the zonal average temperature structure, as well as the shape of the zonal wind jet (Fig. 7a2) is very similar to the unperturbed situation in 2011 (distinct winter stratopause, equatorward tilt of the polar jet). The main differences are that the polar jet is somewhat stronger in the upper statosphere and mesosphere. Further, due to the elongation of the polar vortex (see Fig. 6b3) zonal winds are somewhat stronger already at lower latitudes.



Accordingly, gravity wave squared amplitudes and momentum fluxes (Figs. 7b2 and 7c2) are somewhat stronger at low latitudes in the stratosphere, and momentum fluxes are also somewhat more enhanced around 60° N in the lower mesosphere. Again, high values of gravity wave potential drag are found in the upper part of the polar jet (around the $20\,\mathrm{m\,s^{-1}}$ contour line), where zonal wind vertical gradients are negative. Another enhancement of gravity wave potential drag is located in the

upper mesosphere (above about $80\,\mathrm{km}$ altitude), around the zero wind line. Again, potential drag is comparably weak in the lower part of the polar jet.

### 5.2.3 Shortly before the central date of SSW 2009

During the period 17–21 January 2009, shortly before the central date of the SSW, the thermal structure in the stratosphere is still close to unperturbed conditions (Fig. 7a3). Only the altitude of the polar stratopause is somewhat lower than before.

At latitudes poleward of about 40° N–50° N, however, already the well known mesospheric cooling that is related to SSWs is observed, and the zero wind line has started to descend from about $90\,\mathrm{km}$ altitude down to about $60$–$70\,\mathrm{km}$. On zonal average, eastward directed zonal wind ($u$) has significantly weakened in the whole altitude range. This zonally averaged behavior, however, does not necessarily mean that the polar jet ($\sqrt{u^2+v^2}$) itself has weakened at all altitudes. As can be seen from Figs. 6c2 and 6c3, the polar vortex at $30\,\mathrm{km}$ altitude is still very strong. One reason is the strengthening of the meridional

wind component ($v$). Further, due to the extreme elongation of the vortex, considering a fixed latitude circle the region of strong winds now covers a narrower range of longitudes, resulting in reduced zonal average winds. In addition, larger regions of weak westward winds, apparently secondary circulations and outflows of the polar vortex, have formed at midlatitudes. This indicates that for strongly perturbed vortex conditions a zonal average view may be too simple.

Due to the extreme elongation of the polar vortex, high values of gravity wave squared amplitudes and momentum fluxes

are now spread over the whole Northern Hemisphere (Figs. 7b3 and 7c3). The same is found for gravity wave potential drag (Fig. 7d3). Still, enhanced values of potential drag are found close to the zero wind line at the top of the jet around $85\,\mathrm{km}$ altitude in the latitude range 20° N–40° N, and poleward of 40° N around $70\,\mathrm{km}$ altitude (i.e. somewhat above the zero wind line).

### 5.2.4 Around the central date of SSW 2009

In the period 22–26 January, which is centered around the central date of the SSW, the polar stratopause has further descended, while zonal average zonal winds have further weakened and are westward in the whole altitude range $30$–$90\,\mathrm{km}$ north of 40° N (Fig. 7a4). Gravity wave squared amplitudes and momentum fluxes are still spread out over a large latitude range, but have started to reduce due to the reduction of wind speeds in the split vortex (see Figs. 6d2 and 6d3), and the increased probability of wind reversals at low levels. Also gravity wave potential drag is still spread out over a large latitude range (Fig. 7d4).

Enhancements of potential drag are found above $80\,\mathrm{km}$ altitude and, less pronounced, around $70\,\mathrm{km}$ altitude.





### 5.2.5 Anomalous winds shortly after the central date of SSW 2009

Shortly after the central date, in the period 25–29 January 2009, the polar stratopause has weakened and further descended to about 40 km altitude (Fig. 7a5). At the same time, there are first indications of a new elevated polar stratopause forming at altitudes above 80 km. On zonal average, zonal wind is anomalously westward in the whole Northern Hemisphere in the altitude range 30–50 km, while it is eastward in the altitude range of about 50–85 km equatorward of about 65° N. This, however, is only the zonal average view of the zonal winds. As can be seen from Figs. 6e2 and 6e3, in the stratosphere the longitudinal structure of the horizontal winds is still quite complicated due to the split vortex conditions and the secondary circulations (outflows) of the two polar vortices. Gravity wave squared amplitudes, momentum fluxes and potential drag (Figs. 7b5, 7c5, and 7d5) display a similar zonal average structure as in the previous period (22–26 January), but have further weakened at altitudes below about 60 km.

### 5.2.6 Extended phase of stratospheric anomalous winds

Later during the PJO event, in the period 8-23 February 2009, the old polar stratopause has descended to an altitude of around 20 km, and the new elevated polar stratopause is now well established and has descended to about 75 km (Fig. 7a6). A new strong eastward directed polar jet has formed, which is tilted poleward (with increasing altitude), while zonal winds are still anomalously westward below about 35 km altitude poleward of about 50° N.

Gravity wave squared amplitudes and momentum fluxes are now strongly reduced in the whole Northern Hemisphere with the strongest reduction at latitudes north of 60° N (Figs. 7b6 and 7c6). Still, gravity wave squared amplitudes and momentum fluxes can attain considerable values in the upper part of the newly formed polar jet. A remarkable feature can be seen in the zonal average gravity wave momentum fluxes (Fig. 7c6): In upper stratosphere and lower mesosphere, a broad tongue of enhanced momentum fluxes has formed, which extends from around 35° N at 40 km altitude to around 50° N at 70 km. At the poleward side of this tongue, momentum flux vertical gradients sometimes even reverse, and become positive. This could be an indication for poleward propagation of midlatitude gravity waves into the newly formed strong polar jet.

Gravity wave potential drag (Fig. 7d6) is strongly enhanced at the top of the new polar jet where zonal wind vertical gradients are strongly negative (=westward). This enhanced drag likely contributes significantly to the deceleration of the jet. At the bottom of the new polar jet, however, where zonal wind vertical gradients are strongly positive, potential drag is quite weak. This is the case even though zonal wind vertical gradients at the top and at the bottom of this jet are similarly strong. As already mentioned in Sect. 4.3, this finding is in good agreement with simulations of PJO events by Hitchcock and Shepherd (2013).

Like in most global models, in the simulations of Hitchcock and Shepherd (2013) only purely vertical propagation of gravity waves is assumed. From theoretical considerations, however, refraction of gravity waves into strong wind jets is expected, if full 3D propagation of gravity waves is taken into account (Preusse et al., 2009b; Kalisch et al., 2014). Evidence for this effect from observations has been found, for example, by Jiang et al. (2004b), Ern et al. (2011), or Ern et al. (2013) for gravity waves in the summertime subtropics, and by Hindley et al. (2015) for mountain waves over South America. First indication of gravity wave





meridional propagation for the re-established polar jet during PJO events has been found by Yamashita et al. (2013). These findings can now be further confirmed by the characteristic zonal average distribution of gravity momentum fluxes resulting from our study. As has been pointed out by Yamashita et al. (2013), meridional propagation of gravity waves is usually not considered in gravity wave parameterizations used in global models. For GCMs/CCMs using gravity wave parameterizations that assume only vertical propagation of gravity waves, the simulation of elevated stratopause events and the re-formation of the polar jet after SSWs, as well as the downward propagation of both elevated stratopause and the polar jet after a SSW, may therefore not be fully realistic.

### 5.2.7 Shortly after the period of anomalous winds

Somewhat later during the PJO event, in the period 5–15 March 2009, the new elevated polar stratopause has descended to about 65 km, and also the core of the re-established polar jet has descended to about 60 km (Fig. 7a7).

During this period, gravity wave squared amplitudes and momentum fluxes have started to increase again, particularly at mid latitudes (Figs. 7b7 and 7c7). Likely reason are favorably increased gravity wave saturation amplitudes due to strengthening winds. Further, the probability for gravity waves to encounter wind reversals due to anomalous westward winds has strongly decreased, and troposphere and lower stratosphere will be more permeable for gravity waves than before. Consequently, the tongue of increased momentum fluxes, that was seen during the previous period, is now much less pronounced and broader.

Enhanced values of gravity wave potential drag at the top of the newly formed polar jet are now found at somewhat lower altitudes because meanwhile the jet has somewhat descended in altitude (Fig. 7d7). Further, the absence of a pronounced tongue of enhanced momentum fluxes from lower latitudes seems to indicate that in the later phase of the newly formed polar jet the role of gravity waves propagating meridionally from lower latitudes into the polar jet is less dominant, and the contribution of vertically propagating gravity waves in decelerating the polar jet at its top has increased.

### 6 Gravity waves during different phases of the major SSW in 2006

Now, as a second event, we will investigate the development during different phases of the major SSW / PJO event in winter 2005/2006. We will use the same diagnostics and structure as for the 2009 PJO event. In two ways, this SSW is different from the one in 2009: First, while the SSW 2009 was a split vortex event, dominated by a strong quasi-stationary planetary wave 2, the major SSW in 2006 is a displaced vortex event, dominated by a strong quasi-stationary planetary wave 1. Second, before the central date of the SSW 2009 the zonal average wind in the latitude band 60° N–80° N was strongly eastward, and the wind reversal to westward winds took place on the central date of the major SSW. This is different for the SSW in 2006. On zonal average, the zonal wind in the latitude band 60° N–80° N was oscillating between eastward and westward in the stratosphere well before the central date (21 January 2006). See also Figs. 1e and 2e. Therefore also the evolution of the polar jet before the major SSW 2006, and its effect on the global gravity wave distribution is of interest and will be investigated.





## 6.1 Horizontal distributions during the 2006 major SSW

To illustrate the temporal evolution of the polar vortex in winter 2005/2006, the different rows in Fig. 8 show horizontal distributions at 30 km altitude for different phases of the 2006 PJO event before, around, and after the central date. Shown are ERA-Interim temperatures (left column), ERA-Interim zonal wind (second column) and absolute horizontal wind (third column), as well as HIRDLS gravity wave squared amplitudes (fourth column) and gravity wave absolute momentum fluxes (right column). Because HIRDLS offers a much better horizontal sampling than SABER, horizontal maps of HIRDLS gravity wave activity can be calculated and displayed with much better horizontal resolution.

### 6.1.1 Well before the major SSW 2006, eastward winds around the stratopause

The first period considered is 3–7 January 2006, i.e. well before the major SSW 2006. From Fig. 8a1, it can be seen that already several weeks before the central date of the SSW, there is a pronounced planetary wave 1 structure in the temperature at 30 km altitude. The polar vortex is displaced (Fig. 8a3), but the displacement is not strong enough to result in westward winds on zonal average (Fig. 8a2). Still, there are first indications of vortex instability and breaking of the planetary wave 1: a region of weak westward winds, apparently some outflow out of the polar vortex, is located over North America and the North Pacific, and a region of weak eastward winds, apparently some inflow into the polar vortex, extends from around $30°$ N over the Atlantic Ocean to the Mediterranean.

The bulk of gravity wave activity is found over the North Atlantic, Europe and North Asia, related to the southern part of the displaced vortex. Some enhancement that seems to be related to the weak inflowing and outflowing circulations of the polar vortex is also seen over the Mediterranean, as well as over the North Pacific. Overall, the distribution of gravity wave squared amplitudes and momentum fluxes is quite spotty, which is an indication of strongly intermittent and localized gravity wave sources. Like for the SSW 2009, the observed gravity wave distribution will be a mixture of orographically generated, and of jet-generated gravity waves. The individual sources, however, cannot easily be attributed from the observations alone.

### 6.1.2 Well before the major SSW 2006, westward winds around the stratopause

Somewhat later, during 8–12 January 2006, the zonal average wind in the latitude range $60°$ N–$80°$ N is westward (see Fig. 1e). As can be seen from Fig. 8b1, the phase of the planetary wave 1 has shifted somewhat to the west, and the polar vortex is displaced somewhat more to the south (Fig. 8b3). Due to this slight southward shift, zonal average zonal wind is now westward close to the North Pole (Fig. 8b2).

Although the vortex has only slightly shifted, this has strong effect on the global distribution of gravity wave squared amplitudes and momentum fluxes (Figs. 8b4 and 8b5). The strongest gravity wave activity is no longer found over Northern Asia, but has shifted toward Northern Europe, and strongly increased momentum fluxes are also found over the North Atlantic. The gravity wave activity related to the weak inflow and outflow circulations is somewhat reduced, and there is a shift from the central Mediterranean to the western Mediterranean and to the Canary Islands west of Africa. Overall, these strong changes show the strongly intermittent nature particularly of orographically generated gravity waves (e.g., Eckermann and Preusse,





1999; Jiang et al., 2002; Hertzog et al., 2008, 2012; Wright et al., 2013), but obviously also jet-related gravity wave sources show strong day-to-day variability, as could be the case over the North Atlantic.

### 6.1.3 Around the central date of major SSW 2006

The period 19–23 January is centered around the central date (21 January) of the major SSW 2006. The planetary wave 1 is strongly displaced towards lower latitudes, such that the warm phase of the planetary wave leads to the stratospheric warming in the polar region (Fig. 8c1). The polar vortex is strongly displaced towards Western Europe, and it has considerably weakened (Fig. 8c3). Due to the strong displacement, zonal winds are anomalously westward east of Greenland. Further, winds are slightly westward over North America, Asia, and the North Pacific (Fig. 8c2). Therefore, on zonal average, zonal wind is slightly westward north of about $50°$ N.

Compared to the previous periods, gravity wave activity has considerably weakened. Also compared to the period centered around the central date of the major SSW 2009 gravity wave activity is much weaker. One of the reasons is the much smaller size of the polar vortex during the SSW 2006. This becomes obvious, in particular, when comparing gravity wave momentum flux (5th column) and absolute wind velocities (3rd column) for the two cases in Figs. 6 and 8. Around the central date of the major SSW 2009 almost the whole Northern Hemisphere was affected by the split vortex and its secondary inflow and outflow circulations, resulting in large regions of enhanced gravity wave activity. Different from this, around the central date of the major SSW 2006 we find only several hot spots of gravity wave activity in the vicinity of the much smaller polar vortex (Figs. 8c4 and 8c5).

Two hot spots of strong gravity wave momentum fluxes are located over Greenland and the Alps, and may therefore be caused by strong activity of mountain waves. Another hot spot of momentum fluxes is located over the North Atlantic around $30°$ N and $45°$ N, in a region where strong deceleration of the polar jet is observed. This indicates that these gravity waves could be excited by jet-related sources in the vicinity of the jet exit region. Another enhancement of momentum fluxes is found close to the North African coast at the southern edge of the polar vortex. Because enhanced momentum fluxes persistently show up in this region, this may be an indication of orographic sources, for example the Atlas Mountains.

### 6.1.4 Strong anomalous winds shortly after the central date of major SSW 2006

The next period is from 22–26 January 2006. Somewhat overlapping with the previous period, it covers the phase of strongest stratospheric anomalous westward winds in the polar region (Figs. 8d2 and 8d3). While temperatures in the polar region are still strongly enhanced due to the displaced planetary wave 1 (Fig. 8d1), the polar vortex has further weakened and has started to decay (Fig. 8d3). Gravity wave activity is now strongly reduced (Figs. 8d4 and 8d5). Apart from some scattered gravity wave activity, there are just three momentum flux maxima south of $60°$ N. The first maximum is located over the Alps, the second over the Atlas Mountains, and the third over Eastern Europe. Not much momentum flux is left north of $60°$ N, possibly a consequence of the anomalous westward winds causing wind reversals at lower altitudes.



### 6.1.5 Growing stage of new polar jet

The following period is from 1–10 February 2006. As can be seen from Fig. 1e, this is a phase of anomalous westward winds that are persisting in the polar stratosphere. These winds are situated below a polar jet that is newly forming at higher altitudes. From Fig. 8e1 we find that the planetary wave 1 has almost dissipated, and only little temperature variation is left in the

Northern Hemisphere at 30 km altitude. Also the polar vortex has decayed, and winds at 30 km are anomalously westward in most of the polar region (Figs. 8e2 and 8e3). Enhanced gravity wave activity is found mainly south of 60° N over Western Asia and Northern Africa, related to the final remnants of the polar vortex (Figs. 8e4 and 8e5). Particularly the Western Asian region is very mountainous, and part of the momentum flux enhancement may be due to mountain waves.

### 6.1.6 Mature stage of new polar jet

During 27 February until 3 March 2006, a new temperature minimum has started to form close to the North Pole (Fig. 8f1). Winds in the polar region are still weak, but at mid latitudes three regions of enhanced eastward wind have emerged (Figs. 8f2 and 8f3). The first region is located above the coast of Northeast America and extends over the Atlantic Ocean, the second region is located over the Mediterranean, and the third region is located over Northeast Asia.

Interestingly, strong gravity wave activity is found closely related to those three regions of enhanced winds (Figs. 8f4

and 8f5). While some of the gravity wave activity will be related to favorable enhancements of saturation amplitudes or to sources directly related to those wind systems, also hot spots of gravity wave momentum fluxes are located close to known sources of mountain waves. For example, increased momentum fluxes are seen over North America in the vicinity of the Appalachian Mountains, and along the Eastern Canadian coast. Also the gravity waves around the Mediterranean Sea may partly be mountain waves. The strongest hot spot, however, is located over the coast range of Northeast Asia. Some enhancements

are also seen over Kamchatka and over the mountains in Southeast Russia and Northeast China. Another hot spot located at 40° N, 180° E might be caused by a localized weather system.

### 6.1.7 First weakening of new polar jet

Finally, during the period 10–14 March 2006, the temperature minimum is somewhat displaced from the North Pole (Fig. 8g1). Over most of the Northern Hemisphere, we find weak eastward directed winds. Only over the North Pacific the winds are quite

weak and partly directed westward (Figs. 8g2 and 8g3). The strongest winds are found over North America, the Atlantic Ocean, and close to the North Pole, north of the Bering Strait.

During this period, we find scattered gravity wave activity in most of the Northern Hemisphere, even at latitudes north of 60° N. There is only one major region of low gravity wave activity over the North Pacific. This region coincides with the above mentioned region of comparably weak background winds. Another region of low gravity wave activity is found over the North

Pole. However, compared with the previously discussed period 27 February until 3 March, its area is somewhat smaller, and its gravity wave activity somewhat stronger (cf. Figs. 8f4 and 8g4).




## 6.2 Zonal average cross sections during the 2006 major SSW

In the following, we will investigate zonal average cross sections of MLS temperatures, gravity wave squared amplitudes, absolute momentum fluxes, and gravity wave potential drag during the major SSW in winter 2005/2006 for the same periods as discussed in Sect. 6.1. The gravity wave cross sections are a combination of HIRDLS below 50 km and SABER above

55 km, with a smooth transition between HIRDLS and SABER in the altitude range 50–55 km. The results are given in Fig. 9 for temperatures (upper row), gravity wave squared amplitudes (second row), momentum fluxes (third row), and gravity wave potential drag (bottom row) in the latitude range 20° N–90° N. Again, overplotted contour lines indicate zonal average MLS geostrophic zonal winds.

### 6.2.1 Well before the major SSW 2006, eastward winds around the stratopause

In the first period considered (3–7 January 2006) the polar statopause is not as pronounced as during unperturbed conditions (cf. Figs. 9a1 and 7a1). One of the reasons may be that the stratospheric polar vortex is already somewhat displaced and perturbed (cf. Fig. 8a3). For the same reason, in Fig. 9a1 the zonal average wind in the stratosphere is only slightly positive (=eastward). Also the larger region of weak westward winds that (at 30 km altitude) is located over North America and the North Pacific will contribute to the reduction on zonal average (see Fig. 8a2). Different from this, at 30° N in the mesosphere

one part of the vortex is as strong as about $40\,\mathrm{m\,s^{-1}}$ (i.e., similarly strong as for unperturbed vortex conditions), and therefore does not seem to be affected much.

Gravity wave squared amplitudes and momentum fluxes (Figs. 9b1 and 9c1) are still high with a pronounced maximum around 60° N. As can be seen from Figs. 8a3, 8a4 and 8a5, this maximum is caused by the high gravity wave activity in the southern part of the polar jet. Please note that above 55 km no gravity wave data are available at latitudes north of about 50° N

because SABER is in southward viewing geometry; this is still the case for the time period 8–12 January, discussed in the next subsection. Some enhancement of gravity wave potential drag (Fig. 9d1) is found already in the stratosphere, similar as during unperturbed vortex conditions, or during the period well before the major SSW 2009 (cf. Figs. 7a5 and 7b5). Further, we find a strong enhancement of potential drag close to the top of the mesospheric part of the polar jet.

### 6.2.2 Well before the major SSW 2006, westward winds around the stratopause

During the second period (8–12 January 2006), we find a pronounced polar stratopause (Fig. 9a2). Zonal average zonal wind is negative (=westward) at latitudes north of 40–50° N and altitudes above about 30–40 km. Again, this is an effect of the vortex displacement, and of the large region of weak westward winds over North America and the North Pacific. Compared to the previous period, this region has even grown in size (cf. Fig. 8b2). Like before, the mesospheric part of the polar vortex is strong with zonal average eastward directed wind speeds exceeding $40\,\mathrm{m\,s^{-1}}$ at latitudes south of about 30° N.

Gravity wave activity is enhanced only in the vicinity of the polar vortex and not affected much by the extended region of weak westward winds. Therefore zonal average distributions of squared amplitudes, momentum fluxes and potential drag are very similar to those discussed in Sect. 6.2.1. This shows that a zonal average view may be too simple, if the distribution of





zonal winds has a complicated longitudinal structure. On the other hand, the zonal average distribution of gravity wave activity and potential drag has not changed much, even though there is a considerable change in the longitudinal distribution of gravity wave activity (cf. Figs. 8a5 and 8b5).

### 6.2.3 Around the central date of major SSW 2006

In the time period 19–23 January, centered around the central date (21 January) of the 2006 major SSW, the polar stratopause has started to weaken and to descend (Fig. 9a3). Related to the oscillation of the polar vortex, there were already mesospheric coolings during the periods 3–7 January and 8–12 January (see also Fig. 2e). The cooling around the central date is not more pronounced, and therefore not much difference is seen when comparing Fig. 9a3 with Figs. 9a1 and 9a2. At altitudes above about 30 km the zonal average zonal wind is anomalously westward north of about 50° N. South of 50° N the zonal wind is eastward on average.

In the stratosphere, gravity wave squared amplitudes and momentum fluxes have started to decrease (Figs. 9b3 and 9c3). Still, a maximum is found at latitudes 40-70° N in the stratosphere. This maximum is mainly caused by the gravity wave hot spots seen in Figs. 8c4 and 8c5. Compared to the previous periods, however, gravity wave potential drag is strongly reduced in the stratosphere (Fig. 9d3). In the mesosphere, however, potential drag is strongly enhanced at altitudes above 70 km, where zonal average winds are weakening (or even reversing above 85 km equatorward of 30° N).

### 6.2.4 Strong anomalous winds shortly after the central date of major SSW 2006

Shortly after the central date of the major SSW 2006, during the time period 22-26 January, the polar stratopause has descended to about 40 km, and there is still no indication for the forming of an elevated stratopause (Fig. 9a4). The zonal average distributions of the zonal winds, as well as of the gravity wave squared amplitudes, momentum fluxes and potential drag are very similar to those in the period 19–23 January containing the central date of the SSW. The only differences are that squared amplitudes, momentum fluxes and potential drag in the stratosphere have further decreased (Figs. 9b4, 9c4, and 9d4). In addition, the anomalous westward winds have somewhat descended in altitude, while eastward winds are extending to higher latitudes above about 75 km, thereby forming some kind of transition stage towards the re-establishment of a new poleward tilted eastward directed polar jet.

### 6.2.5 Growing stage of new polar jet

During the period 1–10 February, the old stratopause has descended to below 30 km, and a new stratopause has formed around 75 km (Fig. 9a5). A new eastward directed poleward tilted polar jet has formed, while zonal winds are quite weak and directed westward in the polar stratopause (poleward of 50° N below about 40 km altitude). As a consequence of these weak and westward directed winds, gravity wave squared amplitudes and momentum fluxes are strongly reduced in the polar stratosphere (Figs. 9b5, and 9c5). The momentum flux distribution is somewhat tilted, following the upper part of the new polar jet, but a pronounced tongue of enhanced momentum fluxes, like after the major SSW 2009 (cf. Fig. 7c6), does not show up. Still,





meridional propagation of gravity waves may play an important role. Enhanced gravity wave potential drag is mainly found at the top of the new polar jet (at altitudes above about 60 km) where vertical gradients of the zonal wind are strong (Fig. 9d5). Some weak enhancement of potential drag is also found in the lower part of the new jet in regions of strong vertical gradients of the zonal wind.

### 6.2.6 Mature stage of new polar jet

In the period 27 February until 3 March 2006, the elevated polar stratopause has descended to about 60 km (Fig. 9a6). Compared to the period 1–10 February, the new polar jet has considerably strengthened and descended in altitude. On zonal average, zonal wind in the lower stratosphere is weak and eastward. Gravity wave squared amplitudes and momentum fluxes are still weak in the lower polar stratosphere (Figs. 9b6 and 9c6), but have started to increase around 40 km, related to the increased winds in the polar jet. Still, some poleward tilt is found in the momentum flux distribution at altitudes above about 50 km.

There is some enhancement of gravity wave potential drag in the strong zonal wind vertical gradients at the bottom of the new polar jet (Fig. 9d6). These values are somewhat stronger than those found for comparable conditions during the jet recovery of the 2009 PJO event (Fig. 7d7), which indicates that during the jet recovery of the 2006 PJO event the troposphere and lower stratosphere are more permeable to gravity wave propagation from below. This is confirmed by the horizontal distributions of squared amplitudes and momentum fluxes during these periods (cf. Figs. 6g4, 6g5, and 8f4, 8f5). However, like for the PJO event in 2009, gravity wave potential drag related to the strong zonal wind vertical gradients at the top of the new polar jet is much stronger and likely contributes to the deceleration and reversal of the zonal winds at the top of the jet.

### 6.2.7 First weakening of new polar jet

In the period 10–14 March 2006, the elevated polar stratopause has descended to below 60 km (Fig. 9a7). Also the new polar jet has further descended and has started to weaken. In the stratosphere, gravity wave squared amplitudes and momentum fluxes have further increased (Figs. 9b7 and 9c7), and there is some kind of double peak structure with peaks at about 40° N and 60° N. This double peak is caused by the distribution of the spots of enhanced gravity wave activity, as can be seen from Figs. 8g4, and 8g5. There is only little indication for a poleward tilt of the zonal average momentum flux distribution. The zonal average distribution of gravity wave potential drag (Fig. 9d7) is very similar to the distribution during the previous period of the PJO event. However, potential drag has somewhat decreased in the lower part of the new polar jet, and in the stratosphere potential drag at the poleward side of the jet is now stronger than at the equatorward side.

## 7 Summary and discussion

In our work, we investigate the effect of gravity waves during the boreal winters 2001/2002 until 2013/2014 in the whole middle atmosphere (20–90 km altitude) based on observations of the infrared limb sounding instruments SABER and HIRDLS (depending on data availability).



Altitude–time cross sections illustrate the evolution of zonal average temperatures (Fig. 2) and zonal winds (Fig. 1) at latitudes 60° N–80° N. Temperatures were taken from SABER and MLS observations, as well as from ERA-Interim, and also winds are a composite of ERA-Interim winds and of geostrophic winds derived from MLS and SABER observations. In the thirteen winters considered, there are only a few winters when the polar vortex was little perturbed (the winters 2004/2005, 2010/2011, and 2013/2014), while most of the other winters had at least one major SSW. In six of the perturbed winters, a polar-night jet oscillation (PJO) event took place (in the winters 2003/2004, 2005/2006, 2008/2009, 2009/2010, 2011/2012, and 2012/2013). During these events, the polar stratopause rapidly drops in altitude, a new elevated stratopause forms after the SSW at altitudes around 75 km, and a new polar jet is re-established. Both, elevated stratopause and new polar jet gradually descend in altitude with time over a period of about 1–2 months.

Altitude–time cross sections of observed gravity wave squared amplitudes, momentum fluxes and potential drag (Figs. 3–5) give an overview of the different effects of gravity waves during the different conditions of the polar vortex. Particularly, the interaction between gravity waves and the background winds has strong influence on the gravity wave time series. For example, enhanced values of gravity wave potential drag (i.e., strong vertical gradients of absolute momentum fluxes) are often observed where zonal wind vertical gradients are strong. This is mainly the case in the mesosphere, and related to strong zonal wind vertical gradients at the top of the polar jet, but sometimes enhanced potential drag is also seen together with strong zonal wind vertical gradients related to anomalous westward winds after a SSW.

For unperturbed or just somewhat perturbed vortex conditions, there is notable gravity wave activity already in the stratosphere, because usually there is no wind reversal that would filter out relevant parts of the gravity wave spectrum excited by tropospheric sources. The distribution of gravity wave potential drag indicates that dissipating gravity waves will contribute to the zonal momentum budget in the stratosphere. Values are, however, more enhanced in the upper part of the jet, related to negative (i.e., westward directed) vertical gradients of the zonal wind. This suggests that dissipating gravity waves contribute significantly to the deceleration and reversal of the polar jets in the (upper) mesosphere.

Just before SSWs, sometimes enhanced stratospheric gravity wave activity is found in the latitude range 60° N–80° N. However, this is not always the case. Obviously, the particular shape and position of the polar vortex plays an important role. This is further confirmed by investigating the horizontal distributions of gravity wave activity at 30 km altitude before, around, and after the central dates of the major SSWs 2006 and 2009 (Figs. 6 and 8). Both these SSWs are PJO events, but they are very different in their temporal evolution.

The SSW 2009 is a vortex split event (i.e., strong activity of planetary wave 2). During the evolution of the vortex, we find strong gravity wave activity in regions that are known as hot spots of mountain wave activity, for example over the Rocky Mountains and Scandinavia. However, there is also a lot of gravity wave activity due to jet-related source processes. For example, enhanced gravity wave activity is found in regions of strong jet deceleration (jet exit regions) and of strong curvature of the jet. Of course, the source altitude of the gravity waves seen in these regions at 30 km could be well below this altitude. In addition, enhanced gravity wave activity is found coinciding with patterns of horizontal winds that are caused by secondary circulations (vortex outflows) that seem to be related to the breaking of the planetary wave 2. The gravity wave distribution changes rapidly within only a few days. Further, due to the strong elongation and later split of the vortex, shortly before and





around the central date of the SSW almost the whole Northern Hemisphere north of about 30° N is covered with enhanced gravity wave activity. This is one of the reasons why gravity wave activity increases on zonal average shortly before the central date of the SSW.

The situation is very different for the SSW 2006. This SSW is a vortex displacement event (i.e., strong activity of planetary wave 1). Compared to the SSW 2009, the polar vortex covers a much smaller area, and it even does not elongate much during the evolution of the SSW. Therefore also the area of enhanced gravity wave activity is smaller than during the SSW 2009, and since the vortex does not extend much, gravity wave activity does not increase much on zonal average (at 60° N–80° N) before the central date of the SSW 2006. Also during the evolution of the SSW 2006, there seems to be gravity wave activity due to jet-related sources. However, at 30 km altitude only little activity is found over the patterns of horizontal winds that are apparently caused by secondary circulations (vortex inflow and outflow) related to vortex instability and breaking of the planetary wave 1. On the other hand, there are a lot of very localized hot spots that could be caused by mountain waves, for example over Northeast America, Scandinavia, the Alps, the Atlas Mountains, and later, during the jet recovery, also over Western Asia and Northeast Asia. The distribution of these hot spots changes very rapidly, as a consequence of the rapid vortex evolution, and also due to the strongly intermittent gravity wave source processes.

It should also be mentioned that, for both SSWs considered, the gravity wave distribution follows the absolute wind velocity and displays a strong longitudinal structure. A zonal average view may therefore be too simple, at least during some phases of the vortex evolution. Further, during phases of vortex displacement or elongation, gravity wave activity may be strongly enhanced over a large range of latitudes, which could have an important effect on the overall residual meridional circulation, and thereby on the evolution of the SSW. In particular, our findings support the study by Albers and Birner (2014), and it is suggested that gravity waves may contribute to the triggering of SSWs by preconditioning the shape of the polar vortex such that a SSW can take place.

During PJO events, shortly after the central date of the SSW (or shortly after the SSW in cases of PJO events related to a minor SSW) a new eastward directed polar jet emerges around 75 km altitude. Different from the "regular" polar jet, which is usually tilted equatorward, this newly formed jet is tilted poleward. Below this new jet, zonal winds in the stratosphere are usually very weak, and initially they are prevalently directed anomalously westward. Due to these weak winds, there is no favorable enhancement of gravity wave saturation amplitudes for any gravity wave propagation direction. Furthermore, due to anomalously westward winds, there is an increased probability for low horizontal phase speed gravity waves, for example mountain waves, to encounter critical wind levels.

For these reasons, stratospheric gravity wave activity (amplitudes and momentum fluxes) is very weak during the phase of jet recovery. Therefore, although gravity waves with eastward directed phase speeds in the wide range of about $0$–$80\,\mathrm{m\,s^{-1}}$ encounter critical wind levels in the lower part of the new polar jet, little gravity wave potential drag is observed. Different from this, we find quite strong potential drag in the wind shear at the top of the newly formed polar jet. In spite of the weak stratospheric gravity wave activity, these values of potential drag are comparable to those during unperturbed vortex conditions.

The weak potential drag on the lower flank of the new polar jet indicates that the strong winds in the jet are not caused dynamically by dissipation of eastward propagating gravity waves at high latitudes. Instead, the high wind speeds are likely





the circulation response to the cooling of the polar stratosphere after the SSW, i.e. they are thermally driven. Still, the descent of the shear zone on the upper flank of the new polar jet, and thus also the formation and descent of the newly formed elevated stratopause is likely dynamically driven by breaking gravity waves, as indicated by the enhanced gravity wave potential drag. These findings are qualitatively in good agreement with modeling studies by, for example, Tomikawa et al. (2012), or Hitchcock

and Shepherd (2013).

It is also noteworthy that during the first phase of jet recovery after the SSW 2009, a poleward tilt of the observed zonal average momentum flux distribution indicates that meridional propagation of gravity waves from lower latitudes may also be important for explaining the strong momentum fluxes and potential drag at the top of the new polar jet. This confirms first indications from observed gravity wave variances and a gravity wave ray tracing study by Yamashita et al. (2013). As has

been pointed out by Yamashita et al. (2013), this effect is not included in gravity wave drag parameterizations that assume only vertical propagation of gravity waves. Later during the jet recovery, propagation conditions for gravity waves improve, and vertically propagating gravity waves become more and more important, as indicated by a less tilted momentum flux distribution. During the jet recovery after the SSW 2006, the troposphere and stratosphere seem to be more permeable to gravity waves, and meridional propagation seems to be somewhat less important than during the recovery after the SSW 2009.

Of course, gravity wave observations by limb-viewing satellite instruments such as HIRDLS and SABER have several limitations. First, there are constraints by the observational filter. Only gravity waves with horizontal wavelengths longer than about 100-200 km are visible for those instruments. For details about the observation geometry and the resulting sensitivity for gravity waves see, for example, Preusse et al. (2009a) or Trinh et al. (2015). Second, no directional information is available, and only absolute values of gravity wave momentum flux and potential drag can be derived. Further, errors of observed momentum

fluxes and potential drag are quite large (at least a factor of two).

However, uncertainties in modeling SSWs are sometimes even larger. For example, in model simulations of PJO events values of gravity wave drag at the top of the new polar jet after the SSW range from about $30 \, \mathrm{m \, s^{-1} \, day^{-1}}$ (e.g., Tomikawa et al., 2012; Hitchcock and Shepherd, 2013) to about $150 \, \mathrm{m \, s^{-1} \, day^{-1}}$ (e.g., de Wit et al., 2014). This means that, in spite of their large uncertainties, gravity wave observations by current satellite instruments like HIRDLS and SABER provide an

important confirmation of our physical understanding of the dynamics of SSWs and PJO events. In addition, these observations indicate that models should be improved: The pronounced longitudinal structure and the strong day-to-day variation of the global gravity wave distribution shows the need for global models to include physical gravity wave sources that are as realistic as possible. Further, also non-vertical propagation of gravity waves should be considered. A more quantitative observational approach would be possible by the limb imaging technique, i.e. an improvement of conventional limb measurement techniques

(e.g., Preusse et al., 2009a, 2014; Riese et al., 2014). This technique would provide directional information of gravity waves, and also errors could be considerably reduced.

*Acknowledgements.* This work was partly supported by the Deutsche Forschungsgemeinschaft (DFG) projects PR 919/4–1 (MS–GWaves/SV) and ER 474/3–1 (TigerUC), as well as the Bundesministerium für Bildung und Forschung (BMBF) project no. 01LG1206C (ROMIC/GW–LCYCLE). Work at the Jet Propulsion Laboratory, California Institute of Technology, was done under contract with NASA.



We thank NASA for providing access to the HIRDLS version V006 and to the Aura-MLS version 3.3 level 2 data. These data are freely available via the NASA Goddard Earth Sciences Data and Information Services Center (GES DISC) at http://disc.sci.gsfc.nasa.gov/Aura. ERA-Interim data were obtained from ECMWF (http://www.ecmwf.int). SABER data were provided by GATS Inc. and are freely available at http://saber.gats-inc.com/. The authors would like to thank the teams of the HIRDLS, MLS and SABER instruments for their effort in

5  providing and continuously improving the high-quality data sets used in this study.



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





**Figure 1.** Altitude-time cross-sections of daily 60° N–80° N zonal average zonal wind in $\mathrm{m\,s^{-1}}$ from ERA-Interim, MLS Aura and SABER combined. The time scale on the x-axis is given in "days of the year" (doy) with 1 January 00:00 UT as doy=0. Overplotted contour lines have a contour increment of $20\,\mathrm{m\,s^{-1}}$. Westward (eastward) winds are indicated by dashed (solid) contour lines. The zero wind line is bold solid. Times and altitudes where no data are available are marked in gray.





**Figure 2.** Altitude-time cross-sections of daily 60° N–80° N zonal average temperatures from MLS Aura for the winters 2004/2005–2013/2014, and from ERA-Interim and SABER combined for the winters 2001/2002–2003/2004. The time scale on the x-axis is given in "days of the year" (doy) with 1 January 00:00 UT as doy=0. Like in Fig. 1, overplotted contour lines indicate zonal average zonal winds at 60° N–80° N in steps of $20\,\mathrm{m\,s^{-1}}$.



**Figure 3.** Altitude-time cross-sections of a combination of 1-day zonal average HIRDLS and 3-day zonal average SABER gravity wave squared amplitudes in the latitude band $60°\,N$–$80°\,N$ during the winters 2001/2002–2013/2014, derived in $10\,km$ vertical windows. Time step is $1\,day$ for both HIRDLS and SABER. The time scale on the x-axis is given in "days of the year" (doy) with 1 January 00:00 UT as doy=0. Squared amplitudes are given in $K^2$ on a logarithmic color scale. Like in Fig. 1, overplotted contour lines indicate zonal average zonal winds at $60°\,N$–$80°\,N$ in steps of $20\,m\,s^{-1}$.



**Figure 4.** Same as in Fig. 3, but for gravity wave absolute momentum fluxes at 60° N–80° N. Momentum fluxes are given in mPa on a logarithmic color scale. Like in Fig. 1, overplotted contour lines indicate zonal average zonal winds at 60° N–80° N in steps of $20\,\mathrm{m\,s^{-1}}$.





**Figure 5.** Same as in Fig. 3, but for gravity wave potential drag at 60° N–80° N. Potential drag is given in $\mathrm{m\,s^{-1}\,day^{-1}}$ on a logarithmic color scale. Like in Fig. 1, overplotted contour lines indicate zonal average zonal winds at 60° N–80° N in steps of $20\,\mathrm{m\,s^{-1}}$.





**Figure 6.** Horizontal distributions at 30 km altitude of ERA-Interim temperatures (left column), zonal wind (second column), absolute horizontal wind (third column), and SABER gravity wave squared amplitudes in $K^2$ on a logarithmic scale (fourth column), as well as SABER gravity wave momentum fluxes in mPa on a linear scale (right column). The first row represents an unperturbed vortex situation, averaged over February 13–28 2011, while the other rows represent different periods before, around, and after the central date (24 January) of the 2009 major SSW.







**Figure 7.** Zonal average cross-sections of MLS temperatures (upper), SABER gravity wave squared amplitudes in $K^2$ (second row), momentum fluxes in mPa (third row), and potential drag in $\mathrm{m\,s^{-1}\,day^{-1}}$ (lower) for the latitudes $20°$ N–$90°$ N during different phases of the strong major SSW in winter 2008/2009. For comparison, the left column shows an unperturbed vortex situation during February 2011. For the lower three rows logarithmic color scales are used. Overplotted contour lines indicate the zonal average of MLS geostrophic zonal winds, averaged over the respective time periods shown. Contour interval is $20\,\mathrm{m\,s^{-1}}$. Dashed contour lines indicate westward wind.







**Figure 8.** Horizontal distributions at 30 km altitude of ERA-Interim temperatures (left column), zonal wind (second column), absolute horizontal wind (third column), and HIRDLS gravity wave squared amplitudes in $K^2$ on a logarithmic scale (fourth column), as well as HIRDLS gravity wave momentum fluxes in mPa on a linear scale (right column). The different rows represent different periods before, around, and after the central date (21 January) of the 2006 major SSW.





**Figure 9.** Zonal average cross-sections of MLS temperatures (upper), as well as a combination of HIRDLS and SABER gravity wave squared amplitudes in $\mathrm{K}^2$ (second row), momentum fluxes in $\mathrm{mPa}$ (third row), and potential drag in $\mathrm{m\,s}^{-1}\,\mathrm{day}^{-1}$ (lower) for the latitudes $20^\circ\,\mathrm{N}$–$90^\circ\,\mathrm{N}$ during different phases of the strong major SSW in winter 2005/2006. For the lower three rows logarithmic color scales are used. Overplotted contour lines indicate the zonal average of MLS geostrophic zonal winds, averaged over the respective time periods shown. Contour interval is $20\,\mathrm{m\,s}^{-1}$. Dashed contour lines indicate westward wind.





**Table 1.** Central Dates of Major SSWs in the time period 2001/2002–2013/2014. Vortex displacement events are indicated by "D", and vortex split events by "S". Events with an elevated stratopause forming after the SSW are additionally marked by "ES".

| Winter | Central Date | Type |
|---|---|---|
| 2001/2002 | 2 January | D |
| 2001/2002 | 17 February | D |
| 2002/2003 | 18 January | S |
| 2003/2004 | 5 January | D, ES |
| 2004/2005 | — | — |
| 2005/2006 | 21 January | D, ES |
| 2006/2007 | 24 February | D |
| 2007/2008 | 22 February | D |
| 2008/2009 | 24 January | S, ES |
| 2009/2010 | 9 February | S, ES |
| 2010/2011 | — | — |
| 2011/2012 | — | D, ES |
| 2012/2013 | 7 January | S, ES |
| 2013/2014 | — | — |