# Peer review of "Satellite observations of middle atmosphere gravity wave absolute momentum flux and of its vertical gradient during recent stratospheric warmings"

_Atmospheric Chemistry and Physics, 2016_

## Referee Comment (RC1) · Anonymous Referee #3 · 8 Jun 2016

RECOMMENDATION: minor revision

SUMMARY STATEMENT: The paper is devoted to satelite observations of gravity wave (GW) properties during sudden stratospheric warmings (SSW). It describes in detail GW temperature fluctuations and estimates the absolute momentum flux and its divergence. The findings are interpreted adequately, only the discussion should be clarified in some points and slightly extended. In recognition of the overall quality of the work I suggest: minor revision.

MAJOR COMMENTS:

1) Title: In the title you use the terms "activity and dissipation" which both are well

defined. None of them are observed directly, instead you make estimates of GW momentum flux and its divergence. I suggest to rather refer to "properties" instead.

2) Residual circulation: The basic changes of temperature and wind structure during SSWs can be understood in terms of residual circulation, which is in first instance driven by PWs in the stratosphere and by GWs in the mesosphere. In particular, the zonal-mean zonal wind does not react on the zonal wave drag (as in the QBO) but on the associated poleward flow and vertical motion, inducing dynamical warming and thermal wind changes. As lined out in Hitchcock & Shepherd (2013), there are acting different portions of PWs and GWs on the vertical motion which arranges the wind and temperature fields. When, for example, the GW drag (GWD) appears at the upper flank of the stratospheric eastward jet, the link is through the dynamic warming by breaking westward GWs. In this understanding, the re-formation of the eastward polar jets is not "likely" an effect of the residual circulation, but "basically". Please, see specific comments on page 15 and 16 and remove misunderstandable formulations.

3) Thermal structures: While the aspects of GW generation and propagation are well discussed with relation to the wind, I suggest to add some remarks on the GW impact on the temperature fields. In view of the above, the primary effect of the wave forcings is in a change of residual vertical motion and related dynamical warming. Such patterns should be identified in Fig. 2 and could also be used for a validation of the GW potential drag. In my perception, a part of the GWD in the lower mesosphere is missing which forces the stratopause in the 50 - 70 km altitude range. Please, comment in that.

MINOR COMMENTS

General comments: 1) In the formulae background temperature should be amended with a subscript Zero, as it as done for the background density (see eq. 1, 3) 2) Adjust the timing information between text and figures. For example you refer in page 20, line 26 to 11 - 26 February 2011, but the left column in fig. 7 actually is "119213-110228". There are some more cases like this.

Specific comments: page 1, line 10: The central date is defined as the first day of westward wind. If you refer to the first days after, you should write "when the wind HAS reversed from... (usually AFTER the central...)...". Please, specify. page 8, line 15 and 16: subscribe "0" to background temperature –> $T_0$ page 9, line 19: "before" –> "recently" page 11, line 28: subscribe "0" to background temperature –> $T_0$ page 12, line 23: By definition Polar-night Jet Osciallations (PJO) are based on polar-cap temperatures which is not necessarily associated to planetary wave amplitudes. Insofar ", i.e. when the... maximizes." is not obvious and I suggest to delete this part. page 14, line 12: The amplitude of a conservatively propagating GW does not only depend on density but on wind, too. If you extend the sentence like "... conservatively in a constant wind, this amplitude..." it is correct. page 14, line 14: The conserved quantity is pseudomemoentum flux, which is the same like momentum flux only for mid-frequency GWs. page 15, line 32 (see major comment 2): According to Matsuno, the SSW-related changes in residual circulation in the stratosphere are a result of anomalous breaking planetary waves (PWs). After this perturbation, the polar stratosphere becomes cold again and the PWs re-arrange the normal residual circulation. GWs dominate circulation in the mesosphere and are involved in the formation of apparently downwelling cooling/warming patterns - see Fig. 4 of Hitchcock & Shepherd (2013). Corresponding patterns you see clearly in your temperature plots of Elevated Stratopause (ES) or PJO events, f.e. Fig. 2h. Comparing it with Fig. 5h there is missing the down-reaching cold branch of positive GWD just after the central day. May be it is a matter of the observational data material? The following ES-related warm branch with enhanced negative GWD instead can be identified. page 16, line 6 (see major comment 2): This paragraph is misunderstandable, too. The stratosphere is cold without waves, it is slightly warmed through the PW-driven residual circulation. After the SSW it breaks down and the stratosphere is colder than before. In the mesosphere, the GW-driven branch of residual circulation enforce the stratopause. During SSW, the GWs dissappear (or may change sign) and it becomes cold there. When the winter comes back, GWs rearrange the stratopause. Insofar, the residual circulation drives the thermal structure of the

mesosphere and not vice versa.

REFERENCES:

Hitchcock, P. & T. G. Shepherd, 2013: Zonal-mean dynamics of extended recoveries from stratospheric sudden warmings. J. Atmos. Sci. 70, 2: 688 - 707. doi:10.1175/JAS-D-12-0111.1.

---

## Referee Comment (RC2) · Anonymous Referee #2 · 14 Jun 2016

Review: Satellite observations of middle atmosphere gravity wave activity and dissipation during recent stratospheric warmings, by Ern et al., 2016.

Authors used SABER, HIRDLS, ERA-Interim, and MLS to study gravity wave change during stratospheric sudden warmings (SSWs). This work calculated absolute momentum fluxes and further potential drag that has not yet been intensively studied before. Since it has been difficult to obtain gravity wave drag information globally, this work will provide very interesting results to our community. Authors applied very careful analysis on longitudinal variations of winds, amplitudes, momentum fluxes of gravity waves. Comparisons of gravity waves during two different types of SSWs are interesting. Therefore, I believe that this work is ready for publication after minor revision.

[Figure]
* * *
Interactive
comment

Comments: (1) page 12, lines 28-31. I am confused on these sentences. In this sentences, are you saying that because wind speeds are not strong, so gravity wave sources are probably important. Would you explain a bit more on this? Are you thinking about tropospheric sources? Or jet-related sources? (2) Page 27, lines 20-25. It is interesting. Do you see other anomalies in winds during other SSW events?
* * *

---

## Referee Comment (RC3) · Anonymous Referee #1 · 16 Jun 2016

The authors present an analysis of gravity wave amplitudes, momentum fluxes, and potential drag during 13 Northern hemisphere as estimated from satellite temperature observations from HIRDLS and SABER instruments. They then focus in detail on the three dimensional evolution of these fields during two major warmings/polar night jet oscillation (PJO) events: a split event in early 2009 and a displacement in early 2006.

While the analysis yields only partial information on the gravity wave field (e.g. no directional information regarding the momentum fluxes, and sensitivity to only some parts of the spectrum), the results are nonetheless clearly informative regarding the interaction of the wave field and the background winds. For instance, there is some indication of enhanced wave activity prior to some major warmings, and the wave activity is clearly

**[ACPD](ACPD)**

suppressed during the recovery phase of PJO events, while the drag plays a role in the descent of the newly formed elevated stratopause during the same period, consistent with recent modeling work. There is some evidence as well for meridional propagation during this period which is not captured by current parameterizations.

The analysis is thorough, the paper is clearly written, and these observational results are of clear importance to our understanding of the dynamics of polar night-jet variability, so I am happy to recommend publication with minor revisions. I do have some comments outlined below.

I have four main questions, in no particular order of significance:

a) Role of stratospheric sources

In a number of places the authors seem to imply the possible role of stratospheric sources through jet related generation mechanisms; e.g. p14 l4-7, p18 l10-15, p26 l20-21 p31 l30-32 or by breaking planetary waves p19 l28-29, p31 l33-34. Do the authors believe there to be significant sources within the stratosphere? The possibility that these hotspots are related to selective filtering or to jet-related sources at lower altitudes that are colocated with these features is also raised but I was left feeling unsure of exactly what claims were being made. If these are all simply possibilities being raised that's fine but it would be helpful to have that stated a bit more clearly, perhaps earlier in the discussion. If there are stratospheric sources, might that be an alternative explanation for some of the weak gradients in momentum fluxes seen in Fig. 4 or the apparent meridional propagation in Fig. 7c6?

b) Insensitivity of potential drag to background flow field

p15 l31-32: Presumably this statement is referring to the weak winds in the lower stratosphere (that remain weak well after the winds return to their eastward state)? The winds in the upper stratosphere towards the later part of the recovery phase are quite strong. Do the authors have a sense of whether the weakness of the zonal mean

winds is the key point, or whether the fact that the winds are relatively weak at all longitudes is more relevant? The parameterized orographic fluxes shown in Hitchcock and Shepherd 2013 (their Fig. 5c) recover reasonably quickly once the zonal mean winds return to eastward; this is not so obvious in Fig. 4.

p 17, l4-7: I was a bit surprised that the mesospheric potential drag seems to be relatively insensitive to the background wind field; the presence of westward winds throughout the column doesn't seem to affect the values of the drag in Fig. 5. This again seems at odds with Hitchcock and Shepherd 2013 their Fig. 5a) which shows a strong responses of both orographic and non-orographic parameterized drag to the stratospheric wind reversal. Does this suggest a real difference between the observations and the models?

c) Enhancement prior to sudden warmings

p12 l24-p13 l3: This may to some extend be an issue of the color scale, but it's not so clear to me that the quoted episodes really constitude the periods of highest amplitudes between 30 and 40 km - certainly the peak during the period of westward winds in 2012/13 is the single largest episode, but the periods near day 20 in Fig. 3d, day 0 in Fig. 3f, and much of Figs. 3j and 3b all have comparable amplitudes to my eye. Is the case for enhanced wave activity prior to sudden warmings more clearly made from a zonally asymmetric perspective?

p14 l29-30: It isn't obvious to me why enhanced planetary wave activity should lead to enhanced gravity wave activity - certainly the associated wind field will affect the filtering as is clearly shown in Figs. 6 and 8 but is there evidence that the zonal mean fluxes also increase? Why should this be?

d) Timescale of intermittency

The momentum flux features in both Figs. 6 and 8 are highly localized, and as the authors argue clearly, are related in part to expected filtering processes. Can the authors

comment on the timescale of these hotspots in the satellite observations? Do they typically persist for much of the duration of the averaging periods shown, or do they have shorter timescales?

Finally, one minor point in the introduction:

p3 l13-17; the surface impacts of sudden warmings are not confined to high latitudes - see, e.g Figs.1 and 11 of Hitchcock and Simpson (2014).

P. Hitchcock and I. R. Simpson (2014) 'The Downward Influence of Stratospheric Sudden Warmings' J. Atmos. Sci. 71, 3856-3876 DOI: 10.1175/JAS-D-14-0012.1

---

## Author Comment (AC1) · 20 Jun 2016

Dear Referee #3,

Thank you very much for these very helpful comments!

According to Major Comment #1 we will change the title of the paper to a more appropriate one. Further, we will remove the misunderstandable formulations (mainly on p.15 and 16, see Major Comment #2), and we will add some discussion on the thermal structure (Major Comment #3). Of course, the latter discussion will be more qualitative as it will be only based on observations while some modeling work would be required to understand all processes of relevance, in particular the effect of gravity waves on the

residual circulation.

In addition, we will address the other general and minor comments.

Again, thank you very much for helping us to avoid inaccuracies in the theoretical background and for improving the discussion and interpretation of results!

Sincerely, Manfred Ern

---

## Author Comment (AC2) · 20 Jun 2016

Dear Referee #2,

Thank you very much for your very positive comments!

We will add some more discussion regarding the two minor comments.

Again, thank you very much for you effort!

Sincerely, Manfred Ern

---

## Author Comment (AC3) · 20 Jun 2016

Dear Referee #1,

Thank you very much for thoroughly reading our manuscript and for raising several questions in order to strengthen the discussion of results. We will add some discussion of these questions in the revised manuscript. Please find below a few preliminary comments.

a) about stratospheric sources

Of course, it is not possible to decide from our observations alone whether there are gravity wave sources in the stratosphere. Gravity waves that are observed at 30km can

have their sources at much lower altitudes. In particular, the momentum flux distribution in Fig.7c6 (weak vertical gradients) is likely not caused by stratospheric sources since the weakest or even negative vertical gradients are observed at even higher altitudes (around 55-60km). Therefore meridional propagation of gravity waves is the more likely reason for the weak vertical gradients.

b) about insensitivity of potential drag to the background flow

One of the points mentioned is an apparent mismatch between the time scales of the momentum flux recoveries in our study compared with Fig.5c in Hitchcock and Shepherd (2013). In fact, the time scales of momentum flux recovery are quite comparable; please note that Hitchcock and Shepherd (2013) use a time axis with zero at the central date of the major SSW, while in our figures t=0 is at 1 January 00:00 UT.

This discussion will be included in the revised manuscript. In addition, we will address the other questions raised in the review.

Again, thank you very much for your effort!

Sincerely, Manfred Ern
* * *

---

## Author Response (AR1)

Dear Editor, Dear Reviewers,

Many thanks to the Editor and also many thanks to all Referees for their very helpful comments that significantly helped to improve the manuscript!
Please find included the point-by-point reply to all Reviewer comments. Reviewer comments are given in red color. The reply is given in black, changes made in the manuscript are indicated in blue.

The main concerns by Reviewer #1 can be summarized as follows:
(1) Are there indications for stratospheric gravity wave sources?
(2) Are there differences in gravity wave momentum fluxes and drag comparing Hitchcock and Shepherd (2013) (HS13) with our work?
(temporal evolution of magnitude/time scale of recovery during PJO events)
(3) Is there really an enhancement of gravity wave activity prior to/around SSW central dates?

The main concerns by Reviewer #2 can be summarized as follows:
(4) What could be the reason for enhancements of gravity wave activity prior to/around SSW central dates?

The main concerns by Reviewer #3 can be summarized as follows:
(5) Title of paper is not fully appropriate!
(6) Role of residual circulation for thermal structure and changes of background wind should be discussed more clearly!

These main concerns have been addressed, and the manuscript has been modified as follows:

1. We have added the statement that, from the observations alone, it is difficult to decide whether sources of observed gravity waves are located in the stratosphere.

2. Just after the central date, observed gravity wave momentum flux seems to be stronger than in HS13. This information has been included. Further, a new Fig. 4 has been added in order to show that time scales of recovery in our work are similar as in HS13.

3. From the new Fig. 4, it is indicated that gravity wave squared amplitudes can be enhanced before or around the central date of vortex split events.

4. We have added some reasoning why gravity wave sources may be enhanced for vortex splits.

5. We have changed the title of the paper to:
   "Satellite observations of middle atmosphere gravity wave absolute momentum flux and of its vertical gradient during recent stratospheric warmings"

6. The role of anomalies in the residual circulation is now discussed in more detail in Sect. 4.3. Further, some misleading statements were removed.

For details and our reply to all other comments see the point-by-point reply and the revised manuscript.

Again, thank you very much for your effort!

Sincerely,
Manfred Ern

**1 Reply to Referee #1:**

**1.1 Main Comments:**

**(1) Role of stratospheric sources**

This reviewer comment has been split into two sub-points (1a) and (1b) that are treated separately:

**(1a) In a number of places the authors seem to imply the possible role of stratospheric sources through jet related generation mechanisms; e.g. p14 l4-7, p18 l10-15, p26 l20-21 p31 l30-32 or by breaking planetary waves p19 l28-29, p31 l33-34. Do the authors believe there to be significant sources within the stratosphere? The possibility that these hotspots are related to selective filtering or to jet-related sources at lower altitudes that are colocated with these features is also raised but I was left feeling unsure of exactly what claims were being made. If these are all simply possibilities being raised that's fine but it would be helpful to have that stated a bit more clearly, perhaps earlier in the discussion.**

From our observations alone, it is difficult to obtain information about the source altitude of the observed gravity waves. The source could be at altitudes much lower than the observation altitude. Claims in our manuscript are therefore generally just possibilities being raised.

We have added some discussion of this issue on former p.13, after l.3 (Sect. 4.1.3) when possible effects of gravity waves sources are mentioned for the first time.

**(1b) If there are stratospheric sources, might that be an alternative explanation for some of the weak gradients in momentum fluxes seen in Fig. 4 or the apparent meridional propagation in Fig. 7c6?**

The momentum flux distribution in former Fig.7c6 (weak vertical gradients) is likely not caused by stratospheric sources: the weakest or even negative vertical gradients are observed at altitudes even higher than the stratosphere (around 55–60km). Therefore meridional propagation of gravity waves is the more likely reason for the weak vertical gradients.

We have added the following discussion in Sect. 5.2.6 where former Fig. 7c6 is discussed:

Another explanation for reversed vertical gradients could be gravity wave sources in this altitude range. To our knowledge, however, there are no pronounced gravity wave sources at mid latitudes and altitudes of 50–60 km. Further, it is unlikely that these sources would only be active during one particular time period of a PJO event. Therefore this explanation for reversed momentum flux vertical gradients should be less likely.

**(2) Insensitivity of potential drag to background flow field**
**p15 l31-32: Presumably this statement is referring to the weak winds in the lower stratosphere (that remain weak well after the winds return to their eastward state)? The winds in the upper stratosphere towards the later part of the recovery phase are quite strong.**

Our statement refers to the altitude range of generally weak winds that is located below the re-established polar jet. Of course, there will be some variation in the altitude range that is covered by the weak winds. Particularly in the later phase of jet recovery, weak winds will be confined to the lower stratosphere. This has been clarified by rewording as follows:

One possible explanation for this finding is that, for these situations, background winds in an altitude range below the re-established polar jet are quite weak. At the beginning of the jet recovery weak winds are found in the whole stratosphere, while in the later part of the jet recovery this altitude range covers only the lower stratosphere. As will be seen later in Sects. 5.1 and 6.1, during phases of jet recovery, the zonal wind in the lower stratosphere usually is quite weak at all longitudes. Due to the generally weak winds, also gravity wave saturation amplitudes will be quite low in this altitude range.

The following reviewer comments have been split into three sub-points (2a)–(2c) that are treated separately:

**(2a) Do the authors have a sense of whether the weakness of the zonal mean winds is the key point, or whether the fact that the winds are relatively weak at all longitudes is more relevant?**

It is important that the winds at all longitudes are weak. This has been formulated more clearly in the revised manuscript (see our reply to (2) above).
(If there are localized regions of strong winds, this can lead to localized regions of strong gravity wave activity, resulting in countable zonal averages like during the first half of January 2006.)

**(2b) The parameterized orographic fluxes shown in Hitchcock and Shepherd 2013 (their Fig. 5c) recover reasonably quickly once the zonal mean winds return to eastward; this is not so obvious in Fig. 4.**

There is no obvious mismatch between the time scales of the momentum flux recoveries in our study compared with Fig.5c in Hitchcock and Shepherd (2013). In fact, the time scales of momentum flux recovery are quite comparable; please note that Hitchcock and Shepherd (2013) use a time axis with zero at the central date of the major SSW, while in our figures t=0 is at 1 January 00:00 UT.
In the revised manuscript, we have added another figure (new Fig. 4) that provides gravity wave squared amplitudes for the different winters with a time scale relative to the central date in case a major SSW occurs. From this figure, it can be seen that after a PJO event the recovery of GW activity in the stratosphere at 30-40km altitude has completed around 40–60 days after the central date. This is similar as in Hitchcock and Shepherd (2013): in their Fig.5c the recovery takes place around 40–50 days after the central date.
For changes in the manuscript see the new Figure 4 and related discussion in the revised manuscript (Sect. 4.1.3).

**(2c) p 17, l4-7: I was a bit surprised that the mesospheric potential drag seems to be relatively insensitive to the background wind field; the presence of westward winds throughout the column doesn't seem to affect the values of the drag in Fig. 5. This again seems at odds with Hitchcock and Shepherd 2013 their Fig. 5a) which shows a strong responses of both orographic and non-orographic parameterized drag to the stratospheric wind reversal. Does this suggest a real difference between the observations and the models?**

Thank you very much for pointing this out!
Indeed, this indicates a real difference between observations and Hitchcock and Shepherd (2013), their Fig. 5.
Around the SSW central date and during the phase directly after the central date when winds in the stratosphere and mesosphere are anomalously westward over a large altitude range, the model results by Hitchcock and Shepherd (2013) indicate a strong decrease in gravity wave momentum flux over almost the whole vertical column in the stratosphere and the mesosphere, see their Fig. 5c.
This is somewhat different in the observations: in the observations, during these phases, momentum fluxes are still quite strong in the mesosphere. We find that for all PJO events (see Figs. 5e, 5h, 5i, 5k, and 5l; figure numbering refers to the revised manuscript). This indicates a less effective vertical filtering of gravity waves in the observations. In addition, effects of non-vertical propagation of gravity waves may contribute.
With this not much reduced amount of gravity wave momentum flux still available, the reversed winds after the central date will not lead to a reduction of potential drag in the upper mesosphere.
This discussion has been added in the revised manuscript in Sect. 4.3, point (7).

**(3) Enhancement prior to sudden warmings**

Again, this reviewer comment has been split into two sub-points (3a) and (3b) that are treated separately:

**(3a) p12 l24-p13 l3: This may to some extend be an issue of the color scale, but its not so clear to me that the quoted episodes really constitue the periods of highest amplitudes between 30 and 40 km - certainly the peak during the period of westward winds in 2012/13 is the single largest episode, but the periods near day 20 in Fig. 3d, day 0 in Fig. 3f, and much of Figs. 3j and 3b all have comparable amplitudes to my eye. Is the case for enhanced wave activity prior to sudden warmings more clearly made from a zonally asymmetric perspective?**

For a better illustration of the fact that during vortex split events gravity wave squared amplitudes are enhanced, we have introduced a new Figure 4 showing line plots of gravity wave squared amplitudes. From this figure, it is more clearly seen that for the vortex split events (red curves) gravity wave activity is above the range of 5–10K$^2$ that is found for unperturbed winters (black curves), while gravity wave activity for vortex displacements (blue lines) is similar as for unperturbed vortex conditions.
See Sect. 4.1.3 in the revised manuscript.

**(3b) p14 l29-30: It isn't obvious to me why enhanced planetary wave activity should lead to enhanced gravity wave activity - certainly the associated wind field will affect the filtering as is clearly shown in Figs. 6 and 8 but is there evidence that the zonal mean fluxes also increase? Why should this be?**
Reviewer #1 is correct that our statement on former p14, l29-30 is too general. There is a significant enhancement of gravity wave momentum flux, for example, prior to the major SSW in 2009. However, enhancements of gravity wave momentum fluxes (Fig.5) for other SSWs are not as obvious as for gravity wave squared amplitudes (Fig.3). Figure numbering refers to the revised manuscript.
Therefore the text on former p14 l29-30 has been reworded as follows:
Second, sometimes zonal average gravity wave momentum fluxes are enhanced before or around the central dates of major SSWs, for example for the major SSW in the winter 2008/2009. However, these enhancements are less pronounced than for gravity wave squared amplitudes.
The issue why gravity wave activity can be enhanced during strong planetary wave activity before or around the central dates of SSWs already arises during the discussion of gravity wave squared amplitudes in Sect. 4.1.3. This enhanced gravity wave activity may be related particularly to vortex split events. For a further illustration, we have added the new Fig. 4 and included more discussion in Sect. 4.1.3. For more details see also our reply to comment (1) by Reviewer #2.

**(4) Timescale of intermittency**
**The momentum flux features in both Figs. 6 and 8 are highly localized, and as the authors argue clearly, are related in part to expected filtering processes. Can the authors comment on the timescale of these hotspots in the satellite observations? Do they typically persist for much of the duration of the averaging periods shown, or do they have shorter timescales?**

Of course, there will be strong intermittency of the source regions within the time periods (5 days and longer) that are used for averaging in Figs. 7 and 9 (numbering refers to the revised manuscript). This can be seen by comparing subsequent time intervals in Figs. 7 and 9: the gravity wave distribution can change considerably from one 5-day period to the following.

However, providing reliable information on timescales much shorter than the averaging periods is difficult due to limitations by the daily satellite sampling patterns.

For clarification, we have added the following text at the end of Sect. 6.1.2:
Of course, it is difficult to provide reliable estimates of intermittency time scales because the observations are limited by the daily sampling patterns of the satellite instruments. However, the strong changes from one 5-day period to the following suggest that time scales are much shorter than 5 days. This is further supported by the strong changes in hemispheric gravity wave momentum fluxes by a factor of three from one day to another as obtained from the gravity waves resolved in ECMWF analyses (Preusse et al., 2014).

**(5) Finally, one minor point in the introduction:**
**p3 l13-17; the surface impacts of sudden warmings are not confined to high latitudes - see, e.g Figs.1 and 11 of Hitchcock and Simpson (2014).**
**P. Hitchcock and I. R. Simpson (2014) The Downward Influence of Stratospheric Sudden Warmings J. Atmos. Sci. 71, 3856-3876 DOI: 10.1175/JAS-D-14-0012.1**

This information and the corresponding reference have been included in the revised manuscript.

**2 Reply to Referee #2:**

**2.1 Minor Comments:**

**(1) page 12, lines 28-31. I am confused on these sentences. In this sentences, are you saying that because wind speeds are not strong, so gravity wave sources are probably important. Would you explain a bit more on this? Are you thinking about tropospheric sources? Or jet-related sources?**

Particularly for vortex splits, the vortex may cover a larger area, and the larger area of high wind speeds will generally provide better propagation conditions for gravity waves with phase speeds opposite to the background wind. In addition, jet-related sources may be more active: there could be more jet exit regions than for vortex displacements, and enhanced gravity wave activity is found in the extended regions of vortex inflow or outflow that are related to the dissipation of the planetary wave 2 (more than for a planetary wave 1 event). Further, if the perturbed (split) vortex covers a larger longitudinal range we can expect that more mountain ranges are crossed by the vortex, resulting in an enhanced activity of mountain waves on zonal average.

We have added the following paragraph of discussion in Sect. 4.1.3:

In particular, the three SSWs that had enhanced gravity wave squared amplitudes were all vortex split events (the events in the winters 2008/2009, 2009/2010, and 2012/2013). In Sects. 5.1 and 6.1, we will see that split vortex events may cover a larger longitude range than vortex displacements. On the one hand, this would generally improve propagation conditions for gravity waves propagating opposite to the background wind. On the other hand, this may result in stronger activity of gravity wave sources. For example, there would be more opportunities for the excitation of mountain waves, resulting in enhanced gravity wave squared amplitudes on zonal average. In addition, during vortex split events jet-related gravity wave sources may be more active than during vortex displacements: there could be more jet exit regions, and other jet-related gravity wave source mechanisms could be enhanced, too. The importance of the vortex shape will be discussed in more detail in Sects. 5.1 and 6.1 where gravity wave horizontal distributions in the polar regions are presented.

**(2) Page 27, lines 20-25. It is interesting. Do you see other anomalies in winds during other SSW events?**

This comment refers to the effect of close to zero zonal average zonal wind during non-SSW periods when the zonal wind in the polar vortex is (still) quite strong.

This is an effect that can happen when the polar vortex is shifted off-pole, such that the wind direction is meridional over a range of latitudes, or the vortex may cross the pole such that the wind averages out on zonal average (this happens during the first half of January 2006). The period before the 2006 major SSW is not the only event of this kind. For example, during the period 22-26 January 2010 this is also the case. Therefore the period prior to the major SSW in 2006 is not a singular event, and similar situations can happen from time to time.

The information that the situation during early January 2006 is not a singular event has been added in Sect. 6.2.1 as follows:

A similar situation when the polar vortex is shifted off-pole during a non-SSW period is found during 22-26 January 2010. This suggests that the case from early January 2006 is not a singular event, and that a zonal average view of polar vortex dynamics may be too simple.

In addition, one could think of other situations when, at a fixed latitude, weak secondary inflow and outflow circulations of the polar vortex are opposite to the vortex winds and, on zonal average, can compensate the effect of the polar vortex due to a much larger range of longitudes covered. This effect may however be more prominent at lower latitudes.

**3 Reply to Referee #3:**

**3.1 Main Concerns:**

**(1) Title:**
**In the title you use the terms "activity and dissipation" which both are well defined. None of them are observed directly, instead you make estimates of GW momentum flux and its divergence. I suggest to rather refer to "properties" instead.**

The reviewer is right in that the terms "activity and dissipation" may be misleading; therefore we now refer to GW "absolute momentum flux and its vertical gradient", instead. New title: "Satellite observations of middle atmosphere gravity wave absolute momentum flux and of its vertical gradient during recent stratospheric warmings"

**(2) Residual circulation:**
**The basic changes of temperature and wind structure during SSWs can be understood in terms of residual circulation, which is in first instance driven by PWs in the stratosphere and by GWs in the mesosphere. In particular, the zonal-mean zonal wind does not react on the zonal wave drag (as in the QBO) but on the associated poleward flow and vertical motion, inducing dynamical warming and thermal wind changes. As lined out in Hitchcock & Shepherd (2013), there are acting different portions of PWs and GWs on the vertical motion which arranges the wind and temperature fields. When, for example, the GW drag (GWD) appears at the upper flank of the stratospheric eastward jet, the link is through the dynamic warming by breaking westward GWs. In this understanding, the re-formation of the eastward polar jets is not "likely" an effect of the residual circulation, but "basically". Please, see specific comments on page 15 and 16 and remove misunderstandable formulations.**

In the revised manuscript, Sect. 4.3 point (5) has been partly rewritten by adding a discussion of effects of the residual circulation. Please see our reply to the Specific Comments (8) and (9) below.
In addition, two sentences in the Summary have been rewritten for consistency.

**(3) Thermal structures:**
**While the aspects of GW generation and propagation are well discussed with relation to the wind, I suggest to add some remarks on the GW impact on the temperature fields. In view of the above, the primary effect of the wave forcings is in a change of residual vertical motion and related dynamical warming. Such patterns should be identified in Fig. 2 and could also be used for a validation of the GW potential drag. In my perception, a part of the GWD in the lower mesosphere is missing which forces the stratopause in the 50 - 70 km altitude range. Please, comment in that.**

In addition to the text that was added in reply to Main Concern (2) (see Specific Comments (8) and (9)), we have added some more discussion in the revised manuscript in Sect. 4.3 point (5):
The theoretical picture of the mesospheric gravity wave driven branch of the residual circulation being responsible for changes in the residual vertical motion and related dynamical warming is well supported by the fact that the strongest gravity wave potential drag is usually observed above the temperature maximum of the stratopause (cf. Figs. 2 and 6). The importance of the residual circulation for the formation of the new elevated stratopause is also confirmed by...

Regarding the GWD missing in the lower mesosphere, please see our reply to Specific Comment (9): in Fig. 4 by Hitchcock and Shepherd (2013) the anomaly of GWD is shown, not the net GWD itself. The magnitude of their net GWD is comparably weak in the altitude range and period mentioned (see their Fig.5). The magnitude of our observed potential GWD would allow for even stronger anomalies of net GWD than those shown in Fig. 4 by Hitchcock and Shepherd (2013).

**3.2 Minor Comments**

**(1) In the formulae background temperature should be amended with a subscript Zero, as it as done for the background density (see eq. 1, 3)**

done

**(2) Adjust the timing information between text and figures. For example you refer in page 20, line 26 to 11 - 26 February 2011, but the left column in fig. 7 actually is "119213-110228". There are some more cases like this.**

Thank you very much for noticing these inconsistencies! We have checked the time information in the whole manuscript and corrected, where required.

**3.3 Specific Comments**

**(1) page 1, line 10: The central date is defined as the first day of westward wind. If you refer to the first days after, you should write "when the wind HAS reversed from... (usually AFTER the central...)...". Please, specify.**

Corrected, as recommended.

**(2) page 8, line 15 and 16: subscribe "0" to background temperature $\rightarrow T_0$**

done

**(3) page 9, line 19: "before" $\rightarrow$ "recently"**

done

**(4) page 11, line 28: subscribe "0" to background temperature $\rightarrow T_0$**

done

**(5) page 12, line 23: By definition Polar-night Jet Osciallations (PJO) are based on polar-cap temperatures which is not necessarily associated to planetary wave amplitudes. Insofar ", i.e. when the... maximizes." is not obvious and I suggest to delete this part.**

deleted, as recommended

**(6) page 14, line 12: The amplitude of a conservatively propagating GW does not only depend on density but on wind, too. If you extend the sentence like "... conservatively in a constant wind, this amplitude..." it is correct.**

sentence modified, as recommended

**(7) page 14, line 14: The conserved quantity is pseudomemoentum flux, which is the same like momentum flux only for mid-frequency GWs.**

replaced "momentum" with "pseudomomentum", as recommended

**(8) page 15, line 32 (see major comment 2): According to Matsuno, the SSW-related changes in residual circulation in the stratosphere are a result of anomalous breaking planetary waves (PWs). After this perturbation, the polar stratosphere becomes cold again and the PWs re-arrange the normal residual circulation. GWs dominate circulation in the mesosphere and are involved in the formation of apparently downwelling cooling/warming patterns - see Fig. 4 of Hitchcock & Shepherd (2013). Corresponding patterns you see clearly in your temperature plots of Elevated Stratopause (ES) or PJO events, f.e. Fig. 2h. Comparing it with Fig. 5h there is missing the down-reaching cold branch of positive GWD just after the central day. May be it is a matter of the observational data material? The following ES-related warm branch with enhanced negative GWD instead can be identified.**

In order to point out the role of the residual circulation parts of Sect. 4.3 have been rewritten. For detailed changes see the revised manuscript, as well as our reply to Specific Comment (9).

About the missing branch of positive GWD just after the central day in our former Fig.5h compared with Fig.4b in Hitchcock and Shepherd (2013) (HS13):
Please note that Fig.4b in HS13 shows the *anomaly* of gravity wave drag, not net gravity wave drag! Net gravity wave drag is shown in HS13, their Fig.5. In this figure, the magnitude of net gravity wave drag is strongly reduced just after the central day, and countable values are even shifted to higher altitudes. Comparing our former Fig.5h with Fig.5 in HS13, our values of gravity wave potential drag are even quite strong compared with HS13. This has been pointed out by Reviewer #1 (2c) as being a real difference between HS13 and our former Fig.5. Discussion of this finding has been added in the revised manuscript in Sect. 4.3, point (7).
Given the strong observed potential drag and assuming that the sign of GWD switches at the central date, it might be even possible that the observations indicate a much stronger positive anomaly of gravity wave drag than suggested by Fig.4 in HS13. This speculation has, however, not been added in the revised manuscript because potential drag may differ from net gravity wave drag.

**(9) page 16, line 6 (see major comment 2): This paragraph is misunderstandable, too. The stratosphere is cold without waves, it is slightly warmed through the PW-driven residual circulation. After the SSW it breaks down and the stratosphere is colder than before. In the mesosphere, the GW-driven branch of residual circulation enforce the stratopause. During SSW, the GWs dissapear (or may change sign) and it becomes cold there. When the winter comes back, GWs rearrange the stratopause. Insofar, 
[revised manuscript text omitted]

| 2013/2014 | — | — |